# TRIAGE: Characterizing and auditing training data for improved regression

**Nabeel Seedat**
University of Cambridge
ns741@cam.ac.uk

**Jonathan Crabbé**
University of Cambridge
jc2133@cam.ac.uk

**Zhaozhi Qian**
University of Cambridge
zq224@cam.ac.uk

**Mihaela van der Schaar**
University of Cambridge
mv472@cam.ac.uk

## Abstract

Data quality is crucial for robust machine learning algorithms, with the recent interest in data-centric AI emphasizing the importance of training data characterization. However, current data characterization methods are largely focused on classification settings, with *regression settings* largely understudied. To address this, we introduce TRIAGE, a novel data characterization framework tailored to regression tasks and compatible with a broad class of regressors. TRIAGE utilizes conformal predictive distributions to provide a model-agnostic scoring method, the TRIAGE score. We operationalize the score to analyze individual samples' training dynamics and characterize samples as under-, over-, or well-estimated by the model. We show that TRIAGE's characterization is consistent and highlight its utility to improve performance via data sculpting/filtering, in multiple regression settings. Additionally, beyond sample level, we show TRIAGE enables new approaches to dataset selection and feature acquisition. Overall, TRIAGE highlights the value unlocked by data characterization in real-world regression applications.

## 1 Introduction

**Data characterization, an important problem.** Over a decade ago, it was recognized that "more data beats clever algorithms, but better data beats more data"[1], emphasizing the key role of training data in the performance and robustness of ML algorithms [2–4]. In particular, training data from the "wild" may have various errors or lack coverage, rendering it unsuitable for the ML task [5].

These observations motivate the need for *data characterization*, which aims to score each training sample based on its impact on the performance of the ML task. The training data can then be partitioned or rank ordered according to the score to give a high-level description of the sample and its utility. Data characterization provides a natural way for *data sculpting* [6, 7], which filters training samples that have a low or even detrimental effect on the ML task. The goal is that models built on the sculpted/filtered (and smaller) training data have improved performance.

In this work, we address the *understudied* data characterization problem in **regression settings**. In contrast, existing works on data characterization have primarily focused on the classification setting [8–13]. These methods leverage the probability distribution over discrete labels (softmax or logits) for scoring. However, in the equally important regression setting, the probability distribution over the continuous labels are not typically given by standard regression models. In fact, the majority of the regression models only output a point estimate, e.g. the predicted mean. Without access to probabilistic estimates, one cannot simply apply the existing classification-tailored methods for

37th Conference on Neural Information Processing Systems (NeurIPS 2023).

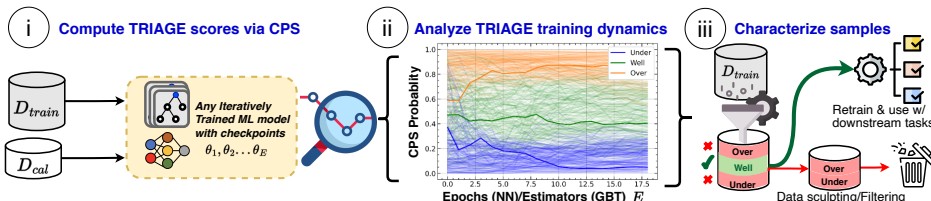

Figure 1: TRIAGE systematically characterizes data in regression settings via a 3-step pipeline. (i) Compute TRIAGE scores: of each sample in the dataset, for each regression checkpoint, via a Conformal Predictive System (CPS). Over the training epochs/iterations, we (ii) Analyze the per-sample training dynamics, studying the trajectory of CPS probability scores for each training sample, with the scoring method operationalized to characterize the samples. (iii) Characterize samples, assigning them to different groups (over-, under- and well-estimated) based on the TRIAGE scores. These groups inform sculpting & other use-cases.

discrete labels to the regression setting. We provide further details why classification-tailored methods are inapplicable to the regression setting in Sec. 2 and Table 1. This non-applicability motivates the need for regression-tailored methods for data characterization, which are "not only principled, but also practical for a larger collection (...) of ML models" [4].

To anchor the need for data characterization in regression, let's consider a few concrete examples. In predicting house prices, an outlier might be a house with an extremely high price due to a unique or under-represented feature in the data. This could affect the model's performance, causing it to overestimate prices for similar houses without the unique feature. Alternatively, imagine predicting patient length of stay based on clinical features. If the dataset contains some erroneous features, the regression predictions may be suboptimal, potentially leading to incorrect clinical decisions.

In designing a data characterization framework for regression, we outline the following desired properties, drawing motivation from the capabilities of works in the classification setting [8–13]:

> **(P1) Consistency:** The characterization of data examples should be stable and consistent across runs and for similar performing models with different parametrizations.
> **(P2) Improve performance:** The characterization should guide the improvements to model performance. For instance, via data sculpting/filtering.
> **(P3) Informative for data collection:** The characterization score should be informative to enable selection between datasets, as well as, feature collection/acquisition.
> **(P4) Regressor versatility:** The characterization should be principled and practical for a large collection of ML models (e.g. not just neural networks).

To fulfill *P1-P4*, we propose a framework to characterize data in regression settings, called **TRIAGE** (s.f. **TR**aining **I**dentification & **A**uditing of **G**ood **E**xamples) — see Fig.1. TRIAGE introduces a **novel regression scoring method**—*TRIAGE score*, building on the recent development of Conformal Predictive Systems (CPS) [14]. CPS is a model-agnostic method that can turn the output of *any* regressor into a probability distribution for continuous labels, termed conformal predictive distribution (CPD). Unlike point estimates such as errors/residuals or conventional conformal prediction intervals, the CPD provides richer information through a full predictive distribution. This enables us to assess the probability of a prediction being over- or under- estimated.

We then operationalize the CPD (and TRIAGE score) in a novel way for data characterization, computing the trajectory of TRIAGE scores over training checkpoints, as a measure of how samples are learned differently during training. Such *training dynamics*, shown in Fig 1 and have been shown to contain a salient signal about the value of the sample to a learner [15–17]. By analyzing the training dynamics, we then categorize training samples as: *under-*, *over-* or *well-estimated* by the model. This is useful going back to our motivating example — where erroneous values could cause the model to overestimate house price. Using the TRIAGE score, we can identify such a sample, as it would likely have a low probability of being correct based on the model's predictive distribution. Such identification unlocks a variety of use cases. For instance, sculpting/filtering such samples can improve model performance (even with less data), as we empirically show in Sec. 5.

**Contributions.** ① *Conceptually*, we present TRIAGE, the first data characterization framework tailored to regression settings, addressing the need for an ML-aware data quality that is principled and practical across a large collection of ML regressors. ② *Technically*, we introduce a principled regression scoring method called TRIAGE score, that leverages conformal predictive distributions in a novel way for data characterization. We operationalize the score to analyze how different samples are

Table 1: Comparing data characterization for classification to TRIAGE (a tailored regression method)

| Type of method | Approaches | Scoring method | Can be repurposed for regression | Applicable to *any* ML model trained iteratively | Reason for non-applicability |
|---|---|---|---|---|---|
| Tailored regression | TRIAGE (Ours) | CPS probability | ✔ | ✔ | - |
| Tailored classification | Data Maps [8] | Model uncertainty | ✗ | ✔ | Needs label probability distribution |
|  | Data-IQ [9] | Data uncertainty | ✗ | ✔ | Needs label probability distribution |
|  | SSFT [10] | Forgetting scores | ✗ | ✔ | Needs discrete class changes |
|  | AUM [12] | Logit margin | ✗ | ✗ | Needs logit output |
| General-purpose | Metadata Archaelogy [18] | Per-sample losses | ✔ | ✔ | - |
|  | VoG [11] | Variance of gradient | ✔ | ✗ | Needs differentiable model (e.g. NN) |
|  | GraNd [13] | Gradient Norm | ✔ | ✗ | Needs differentiable model (e.g. NN) |

learned during training, thereby permitting characterization of individual samples as: *under-*, *over-* or *well-estimated* by the model. ③ *Empirically*, we demonstrate the utility of TRIAGE across multiple use cases satisfying *P1-P4*, including consistent characterization, sculpting to improve performance in a variety of settings, as well as, guiding dataset selection and feature acquisition.

## 2 Related work

This work engages with recent efforts in data-centric AI [6, 7, 19, 20] on data characterization. However, prior methods have focused on classification settings. In contrast, TRIAGE addresses the *regression setting* where tailored data characterization has not been explored. We now discuss related works and outline the challenges of applying prior classification focussed methods to regression, for an overview see Table 1. Prior classification scoring methods can be divided into two groups:

■ **Tailored methods**, assign scores based on aspects specific to classification, including logits [12], probabilities over labels [8, 9] or changes to predictions on discrete labels [10, 21]. However, these methods are inapplicable to our regression setting because these aspects are *not* present in regression.

■ **General purpose methods**, while focussed on classification, they assign scores using generic measures and can be repurposed for regression. However, these general methods based on backpropagated gradients [11, 13] or losses [18] have two limitations. Firstly, scores like gradients can only be used with differentiable models (e.g. neural networks) and are not compatible with many widely used regressors, such as XGBoost — i.e. does not satisfy *(P4) Versatility*. Secondly, even if they can be used, general-purpose scores may not distinguish between samples that have similar magnitude scores (e.g. high loss scores), even if the underlying reasons for these scores may be different with respect to the task. We show the value of such a differentiation between samples in Sec.5.2.

**Data valuation.** A related area to data characterization is that of data valuation, for example: Shapley based valuation methods [22, 23]. We first outline some conceptual differences: (a) TRIAGE unlocks other data-centric tasks beyond just data selection, e.g. feature-level acquisition/collection and selection between datasets. These are out of scope for data "sample" selection; (b) Unlike TRIAGE, these methods are not tailored to regression; (c) method differences: computationally Shapley-based methods need to assess multiple sample permutations and need to retrain the model many times. Hence, they struggle to scale to very high-samples sizes (e.g. 100k), unlike TRIAGE where the cost is cheap comparatively. Recently, a valuation method LAVA [24] has been introduced to address this which uses an additional embedding model to reduce dimensionality before the optimal transport step. Beyond conceptual differences, we compare TRIAGE experimentally and in terms of computation time in Appendix C.

## 3 Data Characterization Preliminaries

In this section, we formulate the data characterization problem for regression settings. Consider the standard regression setting, where the aim is to assign an input $x \in \mathcal{X} \subseteq \mathbb{R}^{d_x}$ to a continuous valued label $y \in \mathbb{R}$ (often a point estimate for $y$). This is done using a regressor $f_\theta : \mathcal{X} \to \mathcal{Y}$, parameterized by $\theta \in \Theta$. We assume $f_\theta$ is trained on a dataset $\mathcal{D}$ with $M \in \mathbb{M}^*$ observations, i.e. $\mathcal{D} = \{(x^m, y^m) \mid m \in [M]\}$. For the purposes of later notation, we consider a sample space $\mathcal{Z} := \mathcal{X} \times \mathbb{R}$, where each observation $z^m = (x^m, y^m) \in \mathcal{Z}$. Typically, parameters $\theta$ are learned in regression, via empirical risk minimization (ERM) with a mean squared error (MSE) loss, i.e. $\text{ERM}(\theta) = \arg\min_{\theta \in \Theta} \frac{1}{M} \sum_{m=1}^{M} \ell(x^m, y^m; \theta)$, with loss function $\ell : \mathcal{X} \times \mathcal{Y} \times \Theta \to \mathbb{R}^+$.

ERM ensures regressor $f_\theta$ performs well *on average*, however it goes without saying that the model is unlikely to perform equally well for all samples in the dataset. Consequently, we desire to characterize such types of samples. Formally, for each sample $x^m$, we aim to assign a *group* label $g^m \in \mathcal{G}$, where

$\mathcal{G} = \{$*Over-estimated, Well-Estimated, Under-estimated*$\}$. We then use these groups for downstream tasks, or to sculpt/filter samples in the dataset $\mathcal{D}$ based on their group assignment. Before giving a precise description of how the group labels are assigned, we provide some context.

Prior works have shown the behavior of samples over the course of model training, called *training dynamics* contains signal about the nature of a sample itself [15–17]. For instance, some samples are easily learned and stable through training, whereas other samples might be challenging and have high variance. Many classification-based methods leverage this concept and assign $g^m$ by analyzing training dynamics e.g. Data-IQ, Data Maps, AUM etc. However, as discussed in Secs. 1 and 2, the aforementioned classification methods *do not* apply to regression. Consequently, this requires us to formalize a new principled scoring method for regression. After that, we can operationalize the scoring method to analyze the training dynamics and assign group labels $g^m$ to each $x^m$.

For regression, such a principled scoring method could evaluate the probability that the true target $y$ is less than or equal to observation $f(x)$, where the categories of samples are: ① *Under-estimated*: $P(y \leq f(x)) \gg 0.5$, ② *Well-estimated*: $P(y \leq f(x)) \sim 0.5$, ③ *Over-estimated*: $P(y \leq f(x)) \ll 0.5$. In the next section, we discuss how we design TRIAGE to achieve this, with the final goal in mind — assigning group labels $g^m$ to each sample $x^m$.

# 4   TRIAGE: characterizing data for regression settings

We now introduce TRIAGE, a regression-focused data characterization framework, compatible with *any* iteratively trained ML regressor (Sec. 4.4). TRIAGE follows a three-step procedure:
1. **Compute TRIAGE scores** (Sec 4.1): Compute the TRIAGE score for every sample $x^m$ using conformal predictive distributions. Do this for every regression checkpoint to obtain a trajectory.
2. **Analyze TRIAGE score training dynamics** (Sec 4.2): Operationalize the TRIAGE score for data characterization by analyzing the TRIAGE score's trajectory for each training sample.
3. **Characterize & assign groups** (Sec 4.3): Characterize samples based on their training dynamics, assigning each sample $x^m$ to a group $g^m$. The groups inform filtering/sculpting or other use cases.

## 4.1   Computing TRIAGE scores via Conformal Predictive Systems (Step 1).

Recall, we want to compute $P(y \leq f(x))$ as a principled scoring method to characterize samples. We want to do so *without* changing the chosen regressor type, i.e. **P4: Regressor versatility**. Of course, the full information of an uncertain label exists in its probability distribution, allowing us to compute probabilities of various outcomes. Unfortunately, regular regressors (*point predictions*) or conventional conformal regressors (*prediction intervals*) do not provide this distribution. While Bayesian methods offer distributions, they require fundamentally changing the regressor.

These challenges motivate defining TRIAGE around Conformal Predictive Systems (CPS)[14]. CPS combines predictive distributions [25, 26] with conformal prediction [27], crucially enabling us to compute the required probability distribution of a continuous variable—called Conformal Predictive Distribution (CPD). CPS also fulfills our versatility goal (i.e. *P4*) as it applies to *any* regressor. To clarify Conformal Predictive Systems output valid cumulative distribution functions, which are termed Conformal Predictive Distributions (CPD). This is the cumulative probability with respect to a label $y$, given some $x$ and regressor $f$. With CPDs denoted as $Q$, the conformal p-values get arranged into a probability distribution which has the properties of a CDF — thus essentially becoming probabilities, see [28]. Since the CPD has the properties of a CDF, we use the CPD to estimate probabilities that the true target is less than or equal to a specified threshold/value.

**Computing CPDs.** TRIAGE formalizes the CPD computation in the framework of the computationally efficient *Split Conformal Predictive Systems* [28]. A *proper training set* $\mathcal{D}_{\text{tr}} = \{(x_{\text{tr}}^i, y_{\text{tr}}^i) = z_{\text{tr}}^i \mid i \in [M]\}$ is used to train regressor $f$. A separate, often smaller, *calibration set* $\mathcal{D}_{\text{cal}} = \{(x_{\text{cal}}^i, y_{\text{cal}}^i) = z_{\text{cal}}^i \mid i \in [q]\}$, is used for conformal calibration. Finally, we define a *conformity measure* $\mu$, where $\mu : \mathcal{Z} \to \mathbb{R} \cup \{-\infty, \infty\}$, quantifies the dataset's agreement with the observation. For each label $y_{\text{cal}}^i \mid i \in [q]$, we compute $q$ conformity scores $\alpha_i = \mu(z_{\text{cal}}^i)$. We define the conformity measure ($\mu$) as per Eq. 1, which is a balanced isotonic conformity measure. We discuss this choice in Appendix A.3, a necessary condition s.t the CPD can be interpreted as a probability distribution [28] [1].

$$\mu(x, y) = {}^{(y-f(x))}\!/\!_{\sigma(x)} \tag{1}$$

---

[1]Under the standard conformal exchangeability assumption (see Appendix A.3), the CPDs are valid.

where $y$ is the label, $f(x)$ a predicted label, $\sigma$ an estimate of prediction quality, computed with a separate normalization model, enabling CPDs to reflect task difficulty. $\sigma$ is kNN fitted on the residuals of $\mathcal{D}_{\text{cal}}$, similar to [29].

We compute the CPD denoted $Q^m$ (Eq. 2) for each sample $x^m$ via Algorithm 1, where the CPD is defined for $y^* \in \mathbb{R}$ [28]. We extract the TRIAGE score using the CPD. We repeat the CPD computation for all samples. Next, we describe our novel approach to operationalize the TRIAGE scores, for the purpose of data characterization.



**Algorithm 1** Computing a CPD

**Require:** Training set $(x_{\text{tr}}^i, y_{\text{tr}}^i) \in \mathcal{D}_{\text{train}}$, $i \in [M]$ and Calibration set $(x_{\text{cal}}^i, y_{\text{cal}}^i) \in \mathcal{D}_{\text{cal}}$, $i \in [q]$.
1: Regressor $f$ is trained on $\mathcal{D}_{\text{train}}$.     ▷ `Training`
2: Compute residuals on $\mathcal{D}_{\text{cal}}$: $R_i = y_{\text{cal}}^i - f(x_{\text{cal}}^i), i \in [q]$
3: Define $\sigma(x)$; which finds the average residuals of the k-nearest neighbors $x$, where KNN is fit on $\mathcal{D}_{\text{cal}}$.
4: Calculate conformity scores $\alpha_1, \alpha_2 ... \alpha_q$ over $\mathcal{D}_{\text{cal}}$ using $\mu$; i.e. $\alpha_i = \mu(x_{\text{cal}}^i, y_{\text{cal}}^i), i \in [q]$.     ▷ `Calibration`
5: Sort $\alpha_1, ... \alpha_q$ in ascending order ; s.t $\alpha_{(1)} < ... < \alpha_{(q)}$
**for** eval sample $x \in \mathbf{X}_{train}$ **do**     ▷ `Evaluation`
    Let $S_{(i)} := f(x) + \alpha_i \sigma(x)$ for $i \in [q]$.
    Let $S_{(0)} := -\infty$ and $S_{(q+1)} := \infty$.
**end for**
Return the CPD as per Equation (2).



$$Q(y^*, \tau) := \begin{cases} \frac{i+\tau}{q+1} & \text{if } y^* \in (S_{(i)}, S_{(i+1)}); i \in [0:q] \\ \frac{i'-1+(i''-i'+2)\tau}{q+1} & \text{if } y^* = S_{(i)}; i \in [1:q] \end{cases} \qquad (2)$$

where $\tau \in [0,1]$, $i' := \min\{j \mid S_{(j)} = S_{(i)}\}$ and $i'' := \max\{j \mid S_{(j)} = S_{(i)}\}$ to account for ties.

## 4.2 Analyzing TRIAGE score trajectories — training dynamics (Step 2)

**Novel use of CPD's for data characterization.** The ***TRIAGE score***, denoted as $T(x, y, \theta) = P(y \leq f_\theta(x))$, represents a novel use of CPDs for data characterization, diverging from their original predictive uncertainty use. Additionally, while CPDs are normally calculated post-model fitting, we observe from Fig.1(ii) that samples' learning varies, leading to different convergence rates of scores. As discussed in Sec. 3, such dynamics reveal sample characteristics [15–17]. Hence, we propose to examine TRIAGE score trajectories to capture the learning dynamics of different samples.

To enable this analysis, we utilize the model checkpoints of *any* ML regressor trained over iterations (e.g. neural networks, XGBoost etc). Specifically, during iterative training, the model parameters $\theta$ vary over $E \in \mathbb{N}^*$ iterations, taking $E$ different parameter values at each checkpoint, i.e. $\theta_1, \theta_2, ..., \theta_E$. We wish to account for the impact of these changing parameters. At each checkpoint $\theta_e$, we compute the CPD to obtain the TRIAGE score for each sample $x^m$. We repeat the computation for every checkpoint to obtain a TRIAGE score trajectory over checkpoints per-sample of dimensionality $E$. i.e. we evaluate the function $Q$ for a specific $f_{\theta_e}(x)$ to get the estimated probability $P(y \leq f_{\theta_e}(x))$. We repeat for all $f_{\theta_e}(x)$ checkpoints where $e \in E$ to get the trajectory of TRIAGE scores for sample.

To quantify and contrast the behavior of the different sample trajectories, we compute two metrics to summarize the per-sample trajectory (see Fig. 2), namely (1) ***Confidence***: reflects the degree of confidence wrt to $P(y \leq f_\theta(x))$. It is computed as the mean of the TRIAGE score over training, i.e. $C(x^i, y^i) = \frac{1}{E} \sum_{e=1}^{E} T(x^i, y^i, \theta_e)$. (2) ***Variability***: reflects the consistency or fluctuation of the score for a specific sample. It is computed as the standard deviation of the TRIAGE score over training, i.e. $V(x^i, y^i) = \sqrt{\frac{\sum_{e=1}^{E}(T(x^i, y^i, \theta_e) - C(x^i, y^i))^2}{E}}$. We describe next how these two metrics form the basis of sample characterization and group assignment.

## 4.3 Characterization & group assignment (Step 3)

We now discuss how the two characterizing metrics, $C$ and $V$, are used to assign a group label $g \in \mathcal{G}$ to each training sample $x^m$. Let's intuitively define each group. ① *Under-estimated (UE)*: low variability $V$ (i.e. decisive about) and HIGH average TRIAGE score $C$, i.e. $P(y \leq f(x)) \gg 0.5$, ② *Over-estimated (OE)*: low variability $V$ (i.e. decisive about) and LOW average TRIAGE score $C$, i.e. $P(y \leq f(x)) \ll 0.5$, and ③ *Well-estimated (WE)*: average TRIAGE score $C$ is uncertain $P(y \leq f(x)) \sim 0.5$. In addition, these samples are often linked to high variability, implying uncertainty about $P(y \leq f(x))$. We empirically observe samples with these different traits per group, depicted in the characteristic curve, see Fig. 2.

We take a threshold-based approach to assign $g^m$, similar to previous characterization works [8, 9]:

$$g^m = \begin{cases} \text{UE} & \text{if } C(x^m, y^m) \geq C_{\text{up}} \wedge V(x^m, y^m) < P_n^{V(\mathcal{D}_{\text{train}})} \\ \text{OE} & \text{if } C(x^m, y^m) \leq C_{\text{low}} \wedge V(x^m, y^m) < P_n^{V(\mathcal{D}_{\text{train}})} \\ \text{WE} & \text{otherwise} \end{cases}$$

(3)

where $C_{\text{up}}$ and $C_{\text{low}}$ are upper and lower confidence threshold resp. and $P_n$ the n-th percentile. We discuss the selection of $C_{\text{up}}$ and $C_{\text{low}}$ in Appendix A.

In Sec. 5, we empirically show how the groups assigned to each sample can be used to sculpt/filter the training dataset or for other downstream tasks such as dataset selection or feature acquisition.

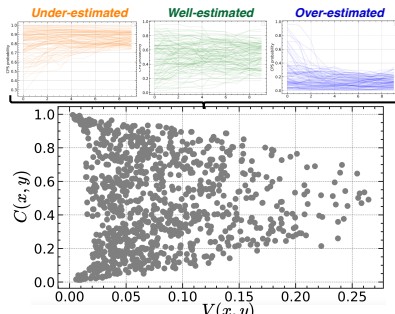

Figure 2: Training dynamics of samples (above) are summarized in a characteristic curve (below-grey)

### 4.4 Using TRIAGE with a variety of regressors (P4)

The *general-purpose* characterization methods discussed are largely compatible with differentiable models (neural networks). However, in real-world settings (e.g., healthcare, finance), practitioners often employ other iterative algorithms, like XGBoost/GBDTs [30]. TRIAGE addresses this limitation, offering a model-agnostic approach using CPS — as the conformity scores can be computed for any regressor. Of course, our use of training dynamics assumes regressor $f$ can be checkpointed. This still confers compatibility with a broader class of models: neural nets, XGBoost, GBDTs, linear regression — *satisfying P4*. We underscore TRIAGE's versatility beyond neural networks, allowing practitioners to use TRIAGE with their preferred model for a specific application. Appendix A.2 details the ease of usage and space and time considerations.

## 5 Empirical investigation

This section presents an empirical evaluation demonstrating TRIAGE [2] satisfies **(P1)** Consistency, **(P2)** Improved regression performance and **(P3)** Informative for data collection. Recall **(P4)** Regressor versatility is satisfied by TRIAGE's design. Additional use-cases and results are in Appendix C.

**Datasets.** We conduct experiments on **10 real-world regression datasets** with varying characteristics. i.e. different sample sizes (500-100k), dimensionality (8-85), and task difficulty. The datasets are drawn from diverse domains, including safety-critical medical regression: (i) Prostate cancer from the US [31] and UK [32], (ii) Hospital Length of Stay [33] and (iii) MIMIC Antibiotics [34]. Additionally, we analyze general UCI regression datasets [35], including Bike, Boston Housing, Bio, Concrete, Protein and Star. The datasets are detailed in Appendix B, along with further experimental details. We observe similar performance across different datasets. However, due to space limitations, we sometimes highlight results for a subset and include results for the remainder in Appendix C.

### 5.1 (P1) Consistent characterization

A key aspect of data characterization is: the ordering of samples matter [9, 10]. This is important when the scores are utilized for data sculpting/filtering. For example, if we rank sort samples by their scores and filter a subset of samples. Consequently, it is crucial to ensure the scoring method used for data characterization is stable and consistent. This consistency is desired across different seeds, but also different practitioners may adopt diverse model architectures/parameterizations for data characterization.

**Consistency of TRIAGE.** We compare the consistency and robustness to variation of TRIAGE vs baseline *general purpose* methods: VoG [11], GradN [13], Loss [18]. We also consider simply computing the end of training Residual Errors (used in the

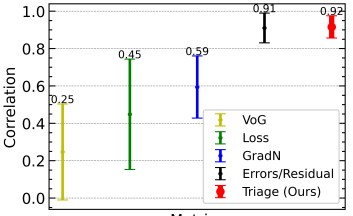

Figure 3: Stability of scoring methods. TRIAGE is most consistent (highest Spearman correlation of *0.91* & lowest variation) averaged across datasets. Results on all 10 datasets are in Appendix C.1.

TRIAGE conformity score). We assess the consistency of the scores across different model architectures/parameterizations (described in Appendix B.3). All models are trained to convergence with early stopping. Similar to prior classification-based characterization methods [9, 10], we assess consistency based on the Spearman rank correlation of the scores evaluated for all model combinations.

Fig. 3, shows TRIAGE is the most consistent compared to baselines, with the highest Spearman correlation (0.91), *satisfying P1*. This consistency means insights derived using TRIAGE scores will also remain consistent. Analyzing each of the 10 datasets (see Appendix C.1), further shows that TRIAGE is stable in magnitude and rank order, whereas the baselines *do not* have such consistency both in magnitude or ordering across datasets, making the baselines challenging to use in practice.

**Takeaway 1.** TRIAGE has the most consistent scoring method compared to baselines over different datasets and model combinations, satisfying P1.

**TRIAGE v Errors/Residuals.** We note that TRIAGE and Errors/Residuals (considered an ablation of TRIAGE's dynamics component) have similar correlations. This raises two questions: *(i) are the samples identified the same?* We stratify the samples in $\mathcal{D}_{\text{train}}$ into three equal sized groups ordered by TRIAGE or Error scores. The overlap is ~0.77, highlighting a difference. This leads us to ask *(ii) what added value does TRIAGE offer?* Consider the case of different samples that have the same error magnitude. Errors/Residuals treat these samples the same, based solely on magnitude. Can TRIAGE offer greater precision to further differentiate samples with the same magnitude?

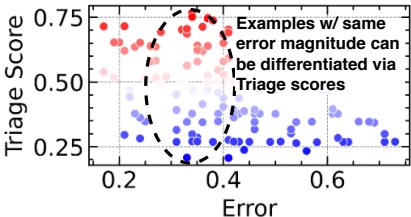

Figure 4: Samples with the same error, have different Triage scores, highlighting the potential to differentiate samples

As shown in Fig.4, TRIAGE scores evidently offer a way to handle samples differently, showing that samples with similar residuals/errors are associated with different TRIAGE scores. We show the phenomena holds across other *general-purpose* scoring rules in Appendix C.10. This result demonstrates the added value of the *training dynamics* aspect, to differentiate samples with the same error magnitude at a more granular level. This contrasts, errors/residuals which are an ablation of TRIAGE, computing the conformity score *only* at the final training checkpoint, instead of for *all* checkpoints. We explore the value of this differentiation to improve regressor performance next.

## 5.2 (P2) Improve regression performance

We now evaluate the ability of TRIAGE to improve regression performance by sculpting/filtering the data — *by keeping only well-estimated samples*. We study two setups using TRIAGE: (1) fine-grained filter (as described above) and (2) sculpting the data based on a deployment purpose.

### 5.2.1 Fine-grained filtering

In the previous section, we showed TRIAGE's ability to differentiate samples with more precision. Does this differentiation with TRIAGE scores benefit model performance? To find out, we sort the training samples from least to most challenging based on the baseline computed scores: Residuals/Errors, Loss, VoG, GradN. We then evaluate test MSE performance, as we increase the proportion $p$ of samples retained for training. i.e. increasingly retain more challenging samples with higher loss/VoG etc. As expected, the baselines (shown in Fig 5, blue) have lower MSE with more sam-

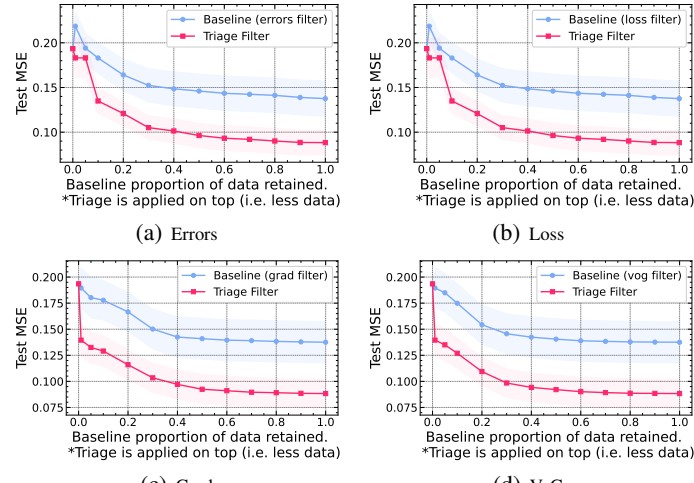

Figure 5: TRIAGE improves test MSE, as a fine-grained filter. For each baseline proportion retained, TRIAGE is applied on top — such that TRIAGE has less data ($D_{Triage} \subset D_{Baseline}$), only keeping well-estimated samples for each proportion $p$.

ples retained for training. However, the elbow around $p = 0.5$, after which MSE does not decrease, shows that the added high-magnitude samples are not beneficial.

We now assess the benefit of differentiating these scores using TRIAGE. To do so, we apply TRIAGE as an additional filter on top of the baseline retained samples — *only keeping the well-estimated subset of samples*. i.e. at each baseline proportion $p$, TRIAGE **reduces the number of samples retained** compared to the baseline s.t. $D_{Triage} \subset D_{Baseline}$. Fig 5 shows, for all proportions $p$, we improve MSE performance over baselines simply by differentiating between samples using TRIAGE and filtering to keep only well-estimated samples. This performance benefit holds for all 4 *general-purpose* scoring methods, illustrating the value of TRIAGE as a fine-grained filter, where even with less data than baselines, the careful selection of samples by TRIAGE leads to improved regression performance. Results on more datasets mirroring the above are in Appendix C.11.

**Takeaway 2.** TRIAGE's differentiation of samples can be used to augment other approaches as a fine-grained filter; improving regressor test performance, simply by keeping well-estimated samples.

### 5.2.2 Data sculpting to fit a deployment purpose.

In many situations, we might want to take a regressor trained on a large $\mathcal{D}_{train}$ and deploy it in an area with limited labeled data. For instance, repurposing a regressor in a low-resource healthcare setting. This raises the question, instead of acquiring new data - which in some cases might be *impossible* - could we sculpt the existing large $\mathcal{D}_{train}$ to be more "fit for purpose". The goal is to enable better regressor performance. Our setup assumes limited access to a small amount of data from the deployment environment, which is *insufficient* in sample size to directly train a highly performant regression model, yet we could use this data for calibration $\mathcal{D}_{cal}$ purposes. Can TRIAGE help?

As a case-study, we consider a regression task similar to [36–38], using patient covariates to predict PSA blood levels, a known predictor of prostate cancer mortality. This task has utility in low-resource areas with limited medical funds [38]. We use multi-country prostate cancer data in which their is a distribution shift and train a regressor on data from the US (SEER)[31]. We evaluate performance on data from the UK (CUTRACT)[32], where the smaller $\mathcal{D}_{cal}$ is also drawn from. In addition to the normal train-large, test-small baseline, we evaluate the utility of sculpting $\mathcal{D}_{train}$ from the US using TRIAGE — *only keeping well-estimated samples* ($D_{WE}$=$D_{Triage}$), s.t $D_{Triage} \subset \mathcal{D}_{train}$. Conformal calibration uses $\mathcal{D}_{cal}$ from the UK.

We compare TRIAGE to several methods, as detailed in Appendix B.3, including: (i) *data-driven* training: training a model directly on the $\mathcal{D}_{cal}$ or combining the two datasets ($\mathcal{D}_{train} \cup \mathcal{D}_{cal}$). (ii) *prediction-based* sculpting: train on $\mathcal{D}_{cal}$ (UK data) and then subsequently filter $\mathcal{D}_{train}$ (US data) based on predictive uncertainty scores [3]. We assess NGBoost, Bayesian Neural Network, Gaussian Process, Bayesian ridge regression, conformal intervals and residuals/errors. We only consider general purpose methods whose criteria allow us to leverage $\mathcal{D}_{cal}$ and are compatible beyond neural nets for flexibility.

We then evaluate MSE test performance of regressors trained on the different "sculpted" $\mathcal{D}_{train}$. We test sensitivity to varying sizes of $\mathcal{D}_{cal}$, as in many settings, we might not have access to many calibration samples. Table 2 shows that TRIAGE consistently outperforms *model-driven* methods across the board. This suggests that predictive uncertainty scores are insufficient to select the samples with the highest utility in $\mathcal{D}_{train}$. This contrast implies TRIAGE's scoring method better selects samples in $\mathcal{D}_{train}$ with greater utility, leading to improved performance on $\mathcal{D}_{test}$.

Compared to *data-driven* approaches, we find that **TRIAGE has the best MSE**, specifically, in the *small sample regime*, where $\mathcal{D}_{cal}$ only contains a limited number of samples $n$=10-200. Of course, when sufficient data is available ($\geq$300 samples), sculpting has a reduced benefit. Rather, training directly on the sufficient $\mathcal{D}_{cal}$ is preferable. The result highlights the viability of sculpting a larger $\mathcal{D}_{train}$ with respect to a small $\mathcal{D}_{cal}$ to improve regression in settings with limited data access — with TRIAGE offering the best performance benefits.

In addition, sculpting especially in healthcare settings should account for both (i) imbalanced feature subgroups and (ii) long-tailed outcomes. We assess this in Appendix C.7 and show that TRIAGE does retains strong performance on minority subgroups and long-tailed outcomes by virtue of the calibration dataset.

---

[3]Model-driven methods are fit on $\mathcal{D}_{cal}$, such that we assess how uncertain samples in $\mathcal{D}_{train}$ are with respect to $\mathcal{D}_{cal}$

Table 2: Comparison of test MSE for different approaches, changing the size of $\mathcal{D}_{\text{cal}}$. TRIAGE's approach to data sculpting of the large $\mathcal{D}_{\text{train}}$ with respect to a small $\mathcal{D}_{\text{cal}}$ in shown to outperform alternatives with lower test MSE, especially when $\mathcal{D}_{\text{cal}}$ from the target domain has small sample sizes. ($\downarrow$ better). Note $|\mathcal{D}_{\text{train}}| \gg |\mathcal{D}_{\text{cal}}|$

| | $\mathcal{D}_{\text{cal}}$ sample size | 10 | 20 | 30 | 40 | 50 | 100 | 200 | 300 | 400 | 500 |
|---|---|---|---|---|---|---|---|---|---|---|---|
| Ours (Sculpting) | **TRIAGE** ($WE$) | **0.051+-0.003** | **0.050+-0.003** | **0.047+-0.002** | **0.046+-0.002** | **0.046+-0.001** | **0.046+-0.002** | **0.046+-0.001** | 0.045+-0.001 | 0.045+-0.002 | 0.046+-0.002 |
| | Not TRIAGE ($OE \cup UE$) | 0.088+-0.014 | 0.068+-0.005 | 0.066+-0.003 | 0.066+-0.005 | 0.068+-0.006 | 0.080+-0.011 | 0.087+-0.015 | 0.093+-0.015 | 0.082+-0.012 | 0.078+-0.009 |
| Data-driven | Baseline ($\mathcal{D}_{\text{train}}$) | 0.064+-0.002 | 0.064+-0.002 | 0.064+-0.002 | 0.064+-0.002 | 0.064+-0.002 | 0.064+-0.002 | 0.064+-0.002 | 0.064+-0.002 | 0.064+-0.002 | 0.064+-0.002 |
| | Baseline ($\mathcal{D}_{\text{cal}}$) | 0.092+-0.023 | 0.070+-0.012 | 0.070+-0.010 | 0.064+-0.003 | 0.066+-0.009 | 0.052+-0.004 | 0.047+-0.001 | **0.043+-0.001** | **0.043+-0.001** | **0.041+-0.001** |
| | Baseline ($\mathcal{D}_{\text{train}} \cup \mathcal{D}_{\text{cal}}$) | 0.060+-0.002 | 0.059+-0.002 | 0.058+-0.002 | 0.056+-0.002 | 0.055+-0.001 | 0.049+-0.001 | 0.047+-0.001 | 0.044+-0.001 | 0.044+-0.001 | 0.042+-0.001 |
| Prediction based sculpting of $\mathcal{D}_{\text{train}}$ | Error Sculpt | 0.066+-0.010 | 0.058+-0.010 | 0.060+-0.010 | 0.060+-0.010 | 0.057+-0.010 | 0.058+-0.010 | 0.055+-0.011 | 0.055+-0.011 | 0.057+-0.010 | 0.057+-0.010 |
| | CP Intervals Sculpt | 0.064+-0.006 | 0.081+-0.023 | 0.082+-0.008 | 0.106+-0.043 | 0.073+-0.013 | 0.059+-0.005 | 0.055+-0.002 | 0.063+-0.011 | 0.054+-0.002 | 0.051+-0.003 |
| | NGBoost | 0.066+-0.006 | 0.121+-0.041 | 0.085+-0.015 | 0.080+-0.008 | 0.094+-0.008 | 0.196+-0.030 | 0.132+-0.010 | 0.096+-0.011 | 0.095+-0.006 | 0.103+-0.008 |
| | Bayesian ridge | 0.080+-0.008 | 0.118+-0.023 | 0.111+-0.012 | 0.129+-0.011 | 0.129+-0.016 | 0.111+-0.017 | 0.106+-0.006 | 0.110+-0.010 | 0.114+-0.009 | 0.103+-0.009 |
| | BNN | 0.068+-0.005 | 0.063+-0.006 | 0.066+-0.004 | 0.066+-0.004 | 0.064+-0.005 | 0.057+-0.006 | 0.072+-0.010 | 0.056+-0.006 | 0.064+-0.003 | 0.068+-0.008 |
| | GP | 0.051+-0.006 | 0.066+-0.008 | 0.069+-0.005 | 0.077+-0.008 | 0.072+-0.006 | 0.071+-0.009 | 0.085+-0.011 | 0.085+-0.003 | 0.095+-0.004 | 0.089+-0.006 |

**Data insights from sculpting.** We seek to understand what types of samples are "sculpted". Such insights are especially useful in clinical settings. We see that US patients ($\mathcal{D}_{\text{train}}$) typically have *higher* cancer stages and *higher* PSA scores than their UK counterparts ($\mathcal{D}_{\text{cal}}/\mathcal{D}_{\text{test}}$). It is precisely these dichotomous samples that are filtered by TRIAGE to improve predictive performance. These insights can be illustrated to stakeholders via a radial diagram, as in Appendix C.5.

Additionally, given the potential data shift (around exchangeability), Appendix C.6 analyzes CPS calibration, along with continuous ranked probability score (CRPS) quantifying the quality of the predictive distribution.

**Beyond predictive performance to fairness.** We also show that sculpting the data has potential beyond simply improving predictive performance. In Appendix C.4, we illustrate how TRIAGE can be used to sculpt data to improve fairness of regressors based on access to a limited set of samples.

**Takeaway 3.** TRIAGE can help to improve regression performance by sculpting a larger dataset to be fit-for-purpose, simply by leveraging a small number of calibration samples.

## 5.3 (P3) Informative for data collection

We now demonstrate the value of TRIAGE for dataset selection and feature collection/acquisition.

### 5.3.1 Dataset selection: From sample-level to dataset level.

In real-world applications, we might want entire dataset characterization, rather than just sample-level characterization. An understudied scenario is how to compare and select between different datasets. This need arises in organizations where data is stored in isolated silos and it is difficult to access; with synthetic data a common solution [39, 40] or alternatively when purchasing data via data marketplaces [41]. We study the former scenario, where we have multiple 'synthetic' data versions, produced by different ML models or data vendors. However, our assessment could easily apply to the latter of data marketplaces.

We explore the setting where the real $\mathcal{D}_{\text{train}}$ is *not accessible* to calculate statistical similarity measures, yet we still wish to compare synthetic datasets and select a version to train on. The underlying assumption is that the synthetic datasets reflect the true underlying distribution. However, depending on the generation process, some synthetic datasets might prove superior to others.

The question we address is: *Can TRIAGE's data characterization allow us to compare and select the synthetic training dataset that yields the best test performance on real data?*

We simulate this scenario by generating synthetic training data using (V1) CTGAN and (V2) TVAE [42], representing 2 synthetic data vendors. We employ TRIAGE to analyze and compare them, based on the percentage of well-estimated examples recommended for retention. As is standard, we train a regressor on synthetic data and test it on real data (TSTR) [43, 44]. The best synthetic data should yield the lowest MAE on real test data. Table 3 shows that where TRIAGE's recommendation has a *higher retention percentage* (i.e. more well-estimated samples), that training with that dataset, leads to lower MAE on real test data. This illustrates that even if the real data is

Table 3: Comparing performance & (Triage retention). Training on synthetic data w/ better utility ($\uparrow$ retained) has the lowest real test data MAE across datasets.

| Dataset | (V1) CTGAN | (V2) TVAE |
|---|---|---|
| Bike | 0.13 (45% Retained) | **0.09 (58% Retained):** |
| Bio | 0.24 (45% Retained) | **0.21(52% Retained)** |
| Boston | 0.16 (38% Retained) | **0.13(60% Retained)** |
| Concrete | 0.19 (46% Retained) | **0.11 (48% Retained)** |
| LOS | 0.16 (47% Retained) | **0.09(56% Retained)** |
| MIMIC | **0.05 (41% Retained)** | 0.08 (48% **Retained**) |
| Prostate | 0.21 (46% Retained) | **0.18 (52% Retained)** |
| Protein | 0.22 (52% Retained) | **0.20 (54% Retained)** |
| Star | 0.18 (38% Retained) | **0.14 (51% Retained)** |

unavailable, TRIAGE's characterization can be used by practitioners to select the synthetic training dataset that will give the best real test performance.

**Takeaway 4.** TRIAGE scores allow us to assess utility between different synthetic datasets in a principled manner. We show that synthetic datasets that TRIAGE suggests retaining *more* samples (i.e. more well-estimated), allow for better generalization when testing on real-data (i.e. MAE).

### 5.3.2 Value of feature collection/acquisition.

Previous settings considered assume a fixed dataset. What if we can actively improve the data's utility by acquiring features? e.g., a doctor runs extra tests to improve a regressor. Understanding the value of features one might acquire is useful, especially if acquisition is costly. We note feature acquisition differs from feature selection (e.g. [45]), where all features are present and we select useful ones. It also differs from active learning, which quantifies sample acquisition value, not features.

We now show that TRIAGE enables a principled approach to assess the benefit of acquiring a specific feature. We posit that acquiring a valuable feature wrt. the task should increase the proportion of well-estimated samples (which we've shown is linked to better performance).

To illustrate the potential of TRIAGE scores, we set up an experiment where we rank the features based on their correlation to the target (i.e. increasing utility). We then simulate the sequential "acquisition" of features with increasing utility (based on correlation). Concurrently, we observe how the groups change as features are acquired.

Fig. 6 shows that as we acquire "valuable" features, the proportion of the well-estimated samples increases, with the over- and under-estimated proportions dropping. This result, repeated on other datasets in Appendix C.12, shows

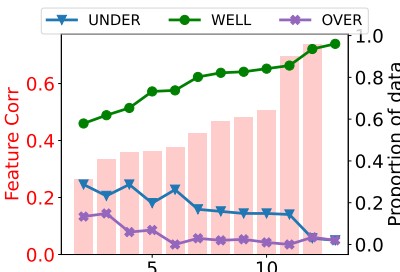

Figure 6: The value of acquiring a feature can be quantified by its ability to increase the proportion of well-estimated samples.

TRIAGE scores are sensitive to the value of a feature, quantified by the increase in the proportion of well-estimated samples.

**Takeaway 5.** TRIAGE offers a principled way to assess active dataset improvement when we acquire new features. The feature's value links to the ability to increase the well-estimated proportion.

## 6 Discussion

We present TRIAGE, a regression-focused data characterization framework, an *understudied* area in data-centric AI. TRIAGE offers a new scoring method to overcome the limitations of classification-centric scores. We show TRIAGE's characterization can unlock a variety of use cases to enhance regression, with minimal effort, beyond standard training on a dataset and exchangeability assumptions. We highlight that a key aspect of ensuring good performance of TRIAGE is careful construction of the calibration set — which should be as representative as possible. Rather than automating or replacing a data scientist's role, TRIAGE as a "data-centric AI" tool aims to empower data scientists to perform the "data" work in a more principled manner.

**Limitations and future opportunities**: (1) understanding the attributes contributing to characterization, (2) provide theoretical guarantees taking into account the challenging interaction between the CPD and Triage trajectory (for more see Appendix C.9), (3) in high-stakes settings, to prevent harm we could leverage domain expertise, where a human-in-the-loop could audit the samples surfaced by TRIAGE, before sculpting.

## Acknowledgements

The authors are grateful to Fergus Imrie, Boris van Breugel, Paulius Rauba and the anonymous NeurIPS reviewers for their useful comments & feedback. The authors would also like to thank Richard Samworth, Marco Scutari, Qingyuan Zhao and Yao Zhang for their insightful discussion on robust statistics linked to TRIAGE. Nabeel Seedat is supported by the Cystic Fibrosis Trust, Jonathan Crabbe by Aviva, Zhaozhi Qian by Cancer Research UK.

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
