# Appendix - TRIAGE: Characterizing and auditing training data for improved regression

## Table of Contents

# A    TRIAGE further details & related work

## A.1    Extended related work

This paper primarily engages with the literature on data characterization and contributes to the nascent area of data-centric AI. We provide specific details of related methods in both these areas below.

**Data characterization.** Existing scoring methods to characterize data examples are primarily for classification tasks. In contrast, our method TRIAGE is to the best of our knowledge the first specifically designed method for regression. As discussed in the main paper, prior classification focussed methods can be divided into two groups. We provide specific details of each below.

**Methods tailored to classification**, these methods rely on aspects only found in classification settings.

- Forgetting scores [21] and Split-Second Forgetting Scores [10] surface difficult and easy examples by analyzing the time when examples change from correct to incorrect discrete class
- AUM [12] identifies mislabeled examples based on logits
- Data Maps [8] and Data-IQ [9] require probabilities of the true class label to distinguish easy, ambiguous, and hard examples.

**General purpose methods**, these methods were built for classification, but could be repurposed for regression.

- Variance of Gradients (VoG) ranks examples by difficulty [11]
- GradN uses gradient norms to identify "important examples" for pruning during training [13]
- Loss values to identify different data subsets [18]
- Residual (Error) analysis, a common statistical approach identifying examples with large model errors.

We argue that methods tailored to classification are not directly related to our regression setting, as they rely on properties specific to classification that are not present in regression. General purpose methods (besides errors & losses) based on their scoring method (e.g. gradients), are limited to differentiable methods like neural networks. Hence, these methods are often inapplicable for example in tabular settings (e.g. healthcare or finance), where practitioners often use iterative learning algorithms like XGBoost. In addition, the general scoring methods are not sufficiently fine-grained to distinguish between examples with the same scores.

Finally, we highlight two specific differences of our approach to all the aforementioned data characterization approaches.

Firstly, all these approaches look to characterize data purely to improve predictive performance. Whilst we also aim to improve predictive performance, we also highlight the value of characterizing data beyond predictive performance with regard to fairness (See Appendix C.4).

Secondly, other methods assume that the model deployment environment is the exactly same as the training environment when characterizing the training data. In contrast, our approach allows for characterization of the training data in a more nuanced manner, with respect to a potential deployment environment with minimal access to a limited amount of data (see Sec 5.2.2).

**Data-Centric AI.** Assessing data quality is a crucial to improve ML system performance, yet is often overlooked in favor of algorithmic development [46]. However, systematic characterization of data could offer guidelines for data enhancement as well as, enable ML system performance improvements [2, 5]. While current approaches when quality is considered is often adhoc or artisinal [7, 19, 46, 47], there has been a recent shift to giving data "center-stage" termed data-centric AI. The goal being to build "systematic methods to evaluate, synthesize, clean and annotate the data used to train and test the AI model" [7, 48].

Our work on characterizing TRIAGE contributes to the best of our knowledge the first method to evaluate data for regression settings, providing an ML-aware data quality monitoring [4].

## A.2 Adapting TRIAGE to any model

### A.2.1 Overview.

TRIAGE's formulation allows us to use it with *any* ML model trained in stages (i.e. iterative learning). Of course, neural networks are easy since we just make use of the model checkpoints. How can we adapt to gradient boosting decision trees (GBDTs) methods - such as XGBoost, LightGBM etc, and statistical methods such as linear regression? As a rule of thumb, we simply need a set of checkpoints through training in order to apply TRIAGE.

This property is incredibly important for practical utility, opening up avenues for more flexible selection of regressors. We look at how TRIAGE can be used with XGBoost and Linear models next.

### A.2.2 XGBoost/GBDTs

**XGB - Overview** Methods such as GBDTs or XGBoost are widely used by practitioners due to their performant nature on tabular data (often outperforming neural networks). By iteratively combininb weak models, it has shown to have great success. Hence, having our method applicable for GBDTs in addition to neural networks adds to the broad utility.

Before outlining why methods such as GBDT's fit the TRIAGE paradigm, we provide a brief overview of GBDTs. Formally, given a dataset $\mathcal{D}_{\text{train}}$, GBDT iteratively constructs a model $F : X \rightarrow \mathbb{R}$ to minimize the empirical risk. At each iteration $e$ the model is updated as:

$$F^{(e)}(x) = F^{(e-1)}(\mathbf{x}) + \epsilon h^{(e)}(\mathbf{x}), \tag{4}$$

where $F^{(e-1)}$ is a model constructed in the previous iteration, $h^{(e)}(\boldsymbol{x}) \in \mathcal{H}$ is a weak learner, and $\epsilon$ is the learning rate. We can consider the model at every $F^{(e)}$ as a checkpoint.

**Space & Time Complexity.** We now provide guidance on how to apply TRIAGE to such methods. Naively, we could construct the checkpoints as an ensemble of multiple independent GBDT's. However, this is inefficient as the space and time complexity scales with $N$ models. To avoid increasing the overhead of a single model (from a practitioner perspective), we create a pseudo-ensemble using a single GBDT, see Figure 7.

**Implementation.** Similar to neural networks, the iterative nature of GBDTs means that the sequential submodels can be considered as checkpoints. Formally, each sub-model has parameters $\theta^{(i)}$, hence the ensemble of checkpoints can be described as $\Theta = \{\theta^{(i)}, 1 \leq i \leq N\}$.

We then apply TRIAGE as normal to the checkpoints $(\theta_1, \theta_2...\theta_E)$. The flexibility of this approach is that it applies both to training a new model, but interestingly, **we can also apply this to an ALREADY trained model by looping through the structure to create the pseudo-ensemble.**

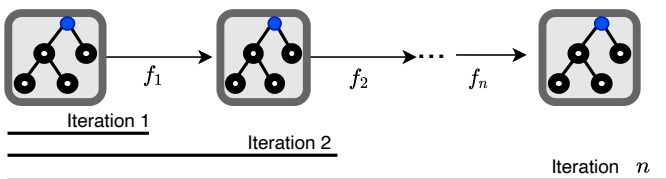

Figure 7: Example illustrating how TRIAGE can be adapted to XGBoost or Gradient Boosting methods by using a pseduo-ensemble. i.e. each sub-model is a checkpoint

### A.2.3 Linear models

**Overview.** Models like linear regression and ridge regression are commonly used statistical models widely used in regression settings. While they are often solved in closed form, we can also learn parameters iteratively. For example, using gradient descent.

**Implementation.** Hence, to use methods like linear regression with TRIAGE, we assume we iteratively learn the parameters via something like gradient descent and store parameters $\theta$ at each iteration, i.e. checkpoint. We can then run TRIAGE as normal.

### A.3    Further details on Conformal Predictive Distributions

This section provides additional details about Conformal Predictive Systems (CPS) [14, 28] that we use in TRIAGE.

Recall CPS combines work from parametric statistics of predictive distributions [25, 26] with the method of conformal prediction [27]. It produces what are called Conformal Predictive Distributions (CPD), estimating the probability distribution of a continuous variable.

We note that this predictive distribution could be transformed to prediction intervals with the corresponding quantiles.

However, in TRIAGE we specifically study the predictive distribution to model the probability of the events related to the data's labels. We provide further details next.

**Split Conformal Predictive System (SCPS).** We provide a formal definition of an SCPS. An SCPS is a function that is BOTH: (i) conformal transducer and (ii) randomized predictive system (RPS).

*(i) Conformal transducer:* As per [14], we thus define a function $Q$ as being a *split conformal transducer*, if $Q$ is determined by a *conformity measure* $\mu$ (the same as defined in the main text)., where $\mu$ measures the degree of agreement between the data set and the observation. Then for each possible label $y_i \in \mathbb{R}$ as part of $\mathcal{D}_{\text{cal}} \in [q]$, we compute $q$ conformity scores $\alpha$. This is the same as described in the main paper.

*(ii) RPS:* The question is, how can a conformal transducer satisfy the properties of an RPS? For more details on this definition of an RPS we refer the reader to [14, 28].

**Motivating the choice of conformity measure.**    The implication that affects TRIAGE is that unlike traditional conformal prediction, where the choice of conformity measure does not matter, in the case of CPS not all conformity measures result in valid predictive distributions.

As shown in [14], in the context of SCPS, the conformity measure $\mu$ must be a balanced isotonic function. See Definition A.1.

**Definition A.1.** A conformity measure $\mu$ is *isotonic* - order preserving if, $y \le y' \Rightarrow \mu(y) \le \mu(y')$

We wish to satisfy this property and apply Proposition 3.1 from [28] that states: "a split conformal transducer based on a balanced isotonic split conformity measure is an RPS".

Hence, this motivates our choice of conformity score, which we define as per Eq. 1 in the main paper.

If $Q$ satisfies (1) a conformal transducer and (2) RPS, then we can relate $Q$ to a CDF of a given $y$, since $Q$ is monotonically increasing in $y \in \mathbb{R}$ and uniformly distributed on [0,1] [14].

*While not critical to the performance of TRIAGE, we wish to highlight a nice property of validity that is provided in TRIAGE through the CPD.*

**Remarks on validity.**    The validity of the CPD is guaranteed if the data is exchangeable between $\mathcal{D}_{\text{cal}}$ and $\mathcal{D}_{\text{test}}$. By validity, this refers to well-calibrated probabilities. (see Assumption A.1). This means that we aren't required to impose any additional requirements for the validity of the CPD, since the aforementioned assumptions on the underlying data are typically made for any ML model.

We further examine the empirical effects on validity under a variety of settings in Appendix C.6.

**Assumption A.1** (Exchangeability). *In a dataset of $n$ observations, the data points do not follow any particular order, i.e., all $n$ permutations are equiprobable. Exchangeability is weaker than IID observations; however, IID observations satisfy exchangeability.*

### A.4    Comparison of Triage to vanilla Conformal Prediction for regression

We wish to highlight some key similarities between Triage and the usage of conventional conformal prediction for regression, e.g. [29, 49].

**Differences.**

- Objective: TRIAGE performs data characterization, scoring samples on their impact on a regressor, enabling data-centric tasks like data sculpting, dataset selection and feature

acquisition. In contrast, [29] is conventional conformal prediction for predictive uncertainty estimation.

- Algorithm: (1) TRIAGE uses conformal predictive distributions (CPDs) providing a full predictive distribution. This contrasts conformal regression's prediction intervals, which are less informative than a predictive distribution. (2) TRIAGE's novelty is studying the training dynamics of scores. In contrast, conformal regression computes prediction intervals once after training, not reflecting dynamic changes, which we show are vital to characterize differences between samples.

- Experiments: TRIAGE tackles data-centric tasks like data sculpting (Sec 5.2), dataset selection (Sec 5.3.1) and feature collection/acquisition (Sec 5.3.2). In contrast, conformal regression like in [29] evaluates the prediction intervals for predictive uncertainty (coverage and efficiency).

- Results: While conformal regression tackles different tasks from TRIAGE, we adapt the CP intervals for sculpting (Sec 5.2.2). Table 2's baseline "CP Intervals Sculpt" corresponds to a conformal regressor. Table 2 shows that porting CP intervals for sculpting are less effective than TRIAGE tailored for this task.

**Similarities:** Similarities are on conformity score and calibration sets usage are general elements fundamental to conformal prediction itself and not unique to any method. This is akin to loss functions (conformity scores) and validation sets (calibration sets).

### A.5 TRIAGE training dynamics

One might wonder why we need to evaluate training dynamics and how it can help us to differentiate between samples.

Of course, at the end of training samples can converge to the same score — however samples are learned differently and at different rates during training. For example: one sample might converge to a CPS probability of 0.9 quickly during training, while another might take longer and oscillate more. Clearly, these samples are not the same.

We show in Figure 8 examples of training dynamics where we have already bucketed the samples as over-, under- and well-estimated. **Each line on the curve represents the training dynamics of a single sample.**

We can clearly see certain samples have much higher varying CPS probabilities over training, whilst others are way more stable.

This motivates our two metrics - Confidence and Variability. Where confidence helps to capture the average trend, but the variability is also important to delineate these oscillating samples.

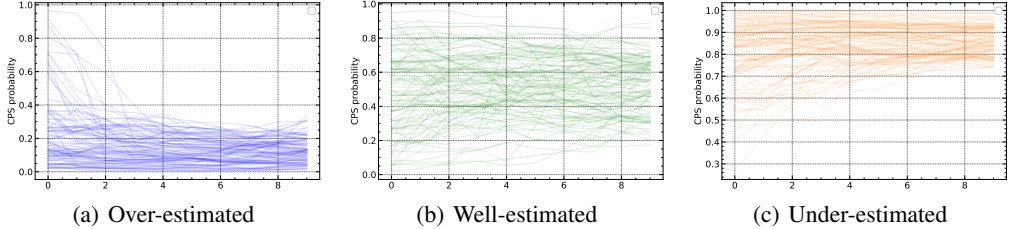

(a) Over-estimated      (b) Well-estimated      (c) Under-estimated

Figure 8: Training dynamics of the different groups – clearly illustrates the necessity and value of training dynamics in capturing the differences between samples. The oscillations also help motivate our two metrics, confidence and variability.

### A.6 TRIAGE thresholds

As outlined in the main text, we stratify samples in a threshold style applied to $C$ and $V$. In particular, the practitioner is required to set $C_{\mathrm{up}}$ and $C_{\mathrm{low}}$. We deem this as a parameter the practitioner is free

to set based on their tolerance levels. For example, someone might consider $> 0.9$ to be confident, whereas another might consider $> 0.7$. These could hence vary from application to application and across problem domains. We take a middle ground such that $C_{\text{up}} = 0.75$ and $C_{\text{low}} = 0.25$.

We however, provide a practical method to guide users in how they might select a threshold for $C_{\text{up}}$ and $C_{\text{low}}$. We define a threshold $thresh \in \{0, 0.5\}$ such that $C_{\text{up}} = 1 - thresh$ and $C_{\text{low}} = thresh$.

Now assume, for any dataset, we train a model and apply TRIAGE. We can then sweep $thresh \in \{0, 0.5\}$ and assess the proportion of examples of well-estimated examples.

We show results in Figure 9 below as we sweep the threshold. For low threshold values (e.g. thresh=0.1, $C_{\text{up}} = 0.9$ and $C_{\text{low}} = 0.1$, where we want to be very certain to filter values. This results in a high proportion of well-estimated examples, which then decreases as the threshold increases.

We observe that our heuristic $C_{\text{up}} = 0.75$ and $C_{\text{low}} = 0.25$, matches the midpoint of this curve, highlighting that this is indeed a reasonable choice

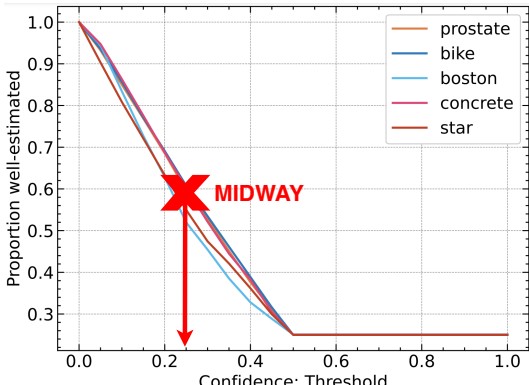

Figure 9: Threshold curve

### A.7  TRIAGE subgroup characterization

With TRIAGE, we can obtain a characteristic curve as a summary of the data where the x-axis represents variability and the y-axis confidence — both metrics computed wrt the TRIAGE score (i.e. CPS probability). We now illustrate where example's lie on the plot.

We highlight in Figure 10, that well-estimated samples are in the middle as they oscillate around 0.5.

We also wish to highlight two types of samples that we DO NOT find in practice. Types of samples we DO NOT find:

- $C >> 0.5$ - very high and also have very high variability $V$
- $C << 0.5$ - very low and also have very high variability $V$

This is likely due to the nature of training dynamics, where samples with very high and very low CPS probabilities are quite clear-cut and hence do not oscillate much over training.

Such phenomena that we observe in practice motivate our rules, which are used to assign samples to the three categories.

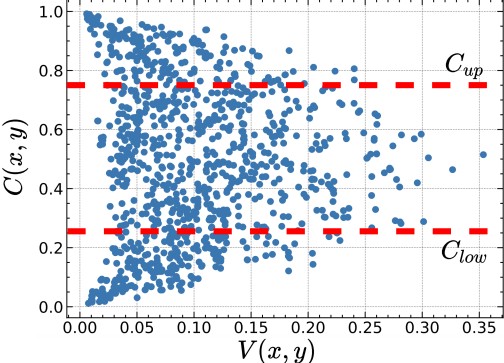

Figure 10: TRIAGE characteristic curve

# B  Benchmarks & Experimental Details

## B.1  Benchmarks

### B.1.1  TRIAGE

The description of TRIAGE is detailed in the main paper, where we compute Conformal Predictive Distributions at each checkpoint $E$ for each training example. We then study the evolution of the CPD through training.

**Implementation details.**  We implement TRIAGE in a versatile manner such that it can plug into regressors built either with Pytorch or Scikit-learn. Our rationale for supporting both types of model API's is that since TRIAGE is applicable to any regressor which we can train iteratively, our functionality should be equally flexible from a usability perspective to foster easy usage and integration.

**Code.**  We will release the code upon acceptance such that TRIAGE can be used by practitioners and researchers alike.

### B.1.2  General-purpose scoring methods

**GraNd**  The gradient norm at epoch $t$ for an input $x, y$ is computed as :

$$\chi_t(x,y) = \mathbb{E} \left\| \sum_{k=1}^{K} \boldsymbol{\nabla}_{f^{(k)}} \ell\left(f_t(x), y\right)^T \psi_t^{(k)}(x) \right\|_2$$

where $\psi_t^{(k)}(x) = \boldsymbol{\nabla}_{\mathbf{w}_t} f_t^{(k)}(x)$.

**Implementation details.** The benchmark is based on [50]. We adapt the Jax implementation from [4] to Pytorch with the help of [5].

**Losses**  Losses to assess training dynamics evaluate the loss at each checkpoint $E$. Such that the score is evaluated on the training trajectory:

$$s_i^e = (l(x_i, y_i, \theta_1), l(x_i, y_i, \theta_2), \ldots, l(x_i, y_i, \theta_E)|(x_i, y_i) \in \mathcal{D}_{\text{train}})$$

**Implementation details.** The benchmark is based on [18] and we use the implementation from [6].

**VoG**  The Variance of Gradient (VoG) is computed across all $E$ checkpoints. At each checkpoint the gradient $G_x$ with respect to input $x$ is computed.

For a given set of $E$ checkpoints, we generate the gradients for each sample $\mathbf{S}$ for all individual checkpoints, i.e., $\{\mathbf{G}_1, \ldots, \mathbf{S}_E\}$. We then calculate the mean gradient $\mu$ by taking the average of the $E$ gradients. The variance of gradients is then:

$$\text{VoG}_{x_i} = \sqrt{\frac{1}{E} \sum_{i=1}^{E} (\mathbf{S}_i - \mu)^2}.$$

**Implementation details.** The benchmark is based on [11] and we use the implementation from [7].

---

[4] https://github.com/mansheej/data_diet
[5] https://github.com/cybertronai/autograd-lib
[6] https://github.com/shoaibahmed/metadata_archaeology
[7] https://github.com/chirag126/VOG

## B.2 Datasets

We provide a summary of the different datasets we use in this paper in Table 4. The datasets vary in number of samples, number of features and domain.

Table 4: Summary of the datasets used in TRIAGE

| Name | $n$ sample magnitude | $n$ features | Domain |
|------|---------------------|--------------|--------|
| Bike [35] | 11k | 18 | General-UCI |
| Bio [35] | 10k | 9 | General-UCI |
| Boston Housing [35] | 500 | 13 | General-UCI |
| Concrete [35] | 1k | 8 | General-UCI |
| CUTRACT Prostate [32] | 2k | 12 | Medical |
| Microsoft Length of Stay [33] | 100k | 23 | Medical |
| MIMIC Antibiotics [34] | 5k | 85 | Medical |
| Protein [35] | 46k | 9 | General-UCI |
| SEER Prostate [31] | 20k | 12 | Medical |
| Star [35] | 20k | 12 | General-UCI |

## B.3 Additional experiment details

We note that all experiments were performed using a single Nvidia Tesla P100 GPU.

### B.3.1 Section 3.1. Robustness to variation experiment details.

We assess the robustness to the variation of different scoring methods when using different models and/or parameterizations.

We parameterize these models differently based on the following changes: (1) Model type, (2) number of layers, (3) number of hidden units, (4) type of optimizer used. We train the models to similar levels of performance for each variant.

We motivate these four changes as these are common changes to model architectures or parameterizations made by practitioners.

We then evaluate all different combinations of these changes over all datasets. We compute both the Spearman correlation for all combinations for each of the 10 datasets.

(1) Model type: Standard MLP and residual MLP (ResMLP) (2) number of layers — we evaluate a 3 layer, 4 layer and 5 layer MLP
(3) number of hidden units — we either reduce the units per layer by 1/2 as we go from the input dimension or by 1/4.
(4) type of optimizer — we assess both Adam and SGD.

Finally, through training, we compute the TRIAGE score and the general purpose scores and then compute the pairwise Spearman correlation of these metrics.

### B.3.2 Section 3.2. Fine-grained filter experiment details.

We train a 5-layer MLP for every regression task. Running TRIAGE and the other general-purpose methods at the end of training to characterize the data (i.e. assign scores to the data). We then take all the general-purpose scores and rank sort them from lowest-highest. Which represents easiest to hardest samples. Thereafter, as a baseline for each method we sweep and retrain a model with an increasing proportion of data included from lowest to highest. As a fine-grained filter, we apply TRIAGE on top of these other methods and show we can improve performance, simply by virtue of differentiating samples. What is interesting is that at each proportion, the TRIAGE version contains less samples.

### B.3.3 Section 3.3. Fit for purpose experiment details.

For this experiment we train a baseline XGBoost regressor model on SEER (US) - $\mathcal{D}_{\text{train}}$. The XGBoost implementation uses the Python version from [8].

We then draw $\mathcal{D}_{\text{cal}}$ from CUTRACT (UK), as is $\mathcal{D}_{\text{test}}$ from the UK but a disjoint set. We then assess the following performances evaluated on the test set wrt. MSE performance. Besides the sculpting and training directly on different subsets of the data, we assess the following model-driven approaches described next.

The following models are trained on $\mathcal{D}_{\text{cal}}$, they are then applied to $\mathcal{D}_{\text{train}}$, where $\mathcal{D}_{\text{train}}$ is then sculpted based on the predictive uncertainty. The following models are assessed: NGBoost [9], Bayesian Neural Network [10], Gaussian Process [11], Bayesian Ridge Regression [12].

### B.3.4 Section 3.4. Beyond example-level to dataset level experiment details.

We compare two synthetic data models as representations of comparing two sources producing synthetic data (e.g. two companies producing synthetic data). We compare CTGAN and TVAE and use their implementations from [13].

### B.3.5 Section 3.5. Active improvement via acquisition experiment details.

We assess the viability of using TRIAGE to quantify the value of acquiring features based on the increase in well-estimated samples. The value in the simulated setup is quantified by the Pearson correlation of the feature to the label. We rank sort these correlations from low to high and sequentially acquire them, running TRIAGE at each instantiation. Our underlying model in which to do the quantification is an XGBoost with 20 estimators.

---

[8] https://github.com/dmlc/xgboost

[9] https://github.com/stanfordmlgroup/ngboost

[10] https://github.com/IBM/UQ360

[11] https://scikit-learn.org/

[12] https://scikit-learn.org/

[13] https://github.com/sdv-dev/SDV

# C    Additional experiments

## C.1    Consistency across datasets

**Goal.**    In the main paper we looked at consistency and stability of the different scoring methods when averaged across the different datasets. We now assess the individual correlations for each dataset in Figure 11.

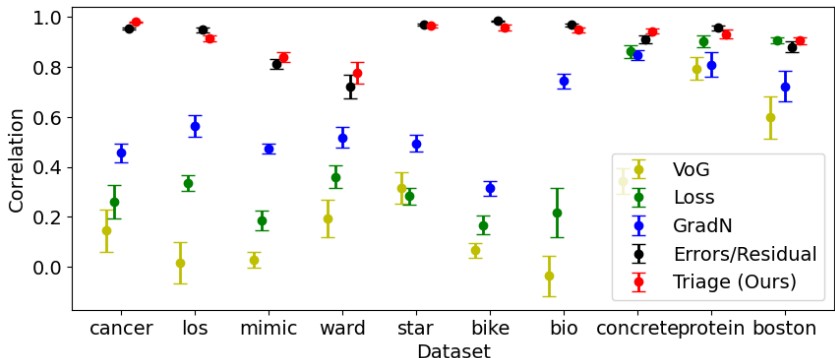

Figure 11: Consistency all datasets

**Takeaway.**    TRIAGE has the highest consistency across all the different datasets compared to the other methods. We also see the baselines do not have consistent performance ordering across datasets, making it challenging to select an appropriate scoring rule.

## C.2    Computation time

**Goal.**    An important question is how does TRIAGE scale to the size of the dataset from a computation time persepctive. To assess this we vary the dataset size from 2000-100k samples and assess the TRIAGE computation time.

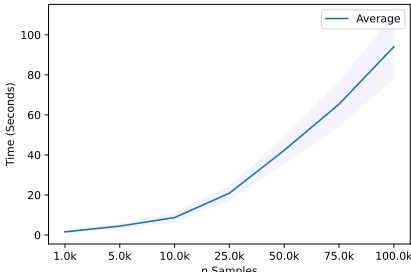

Figure 12: Overall computation time with data sizes varying between 2000-100k samples

**Takeaway.**    As shown in Figure 12, naturally as the data size increases, so does computational time. However, even at 100k samples computing the TRIAGE scores for all 100k takes less than 2 min highlighting TRIAGE's time efficiency to scale to larger data sizes. This suggests that TRIAGE can be used efficiently with larger datasets.

## C.3    TRIAGE w/ other iterative algorithms: XGBoost & CatBoost

**Goal.**    As discussed in the main paper, TRIAGE can be used with any regressor which can be trained iteratively. Methods such XGBoost and CatBoost methods are widely used regressors by practitioners on tabular data, often more so than neural networks [30]. As mentioned, we desire consistency of data characterization, such that samples are consistently characterized across similar performing models irrespective of the model.

We have assessed the neural network consistency, but not assessed this across models such as XGBoost or CatBoost. We train both to achieve a similar level of performance and then apply TRIAGE as an

assessment for these methods. The evaluation of both on different datasets is in Figure 13, where we show characteristic curves.

*Note, the other baselines could not be easily assessed since the models under evaluation are not differentiable.*

**Takeaway.** We can see TRIAGE characterizes samples consistently across the two models, as can be seen across the characteristic curves. We also compute the Spearman correlation of scores across models in Table 5. The mean Spearman correlation across datasets is $0.88 + -0.04$.

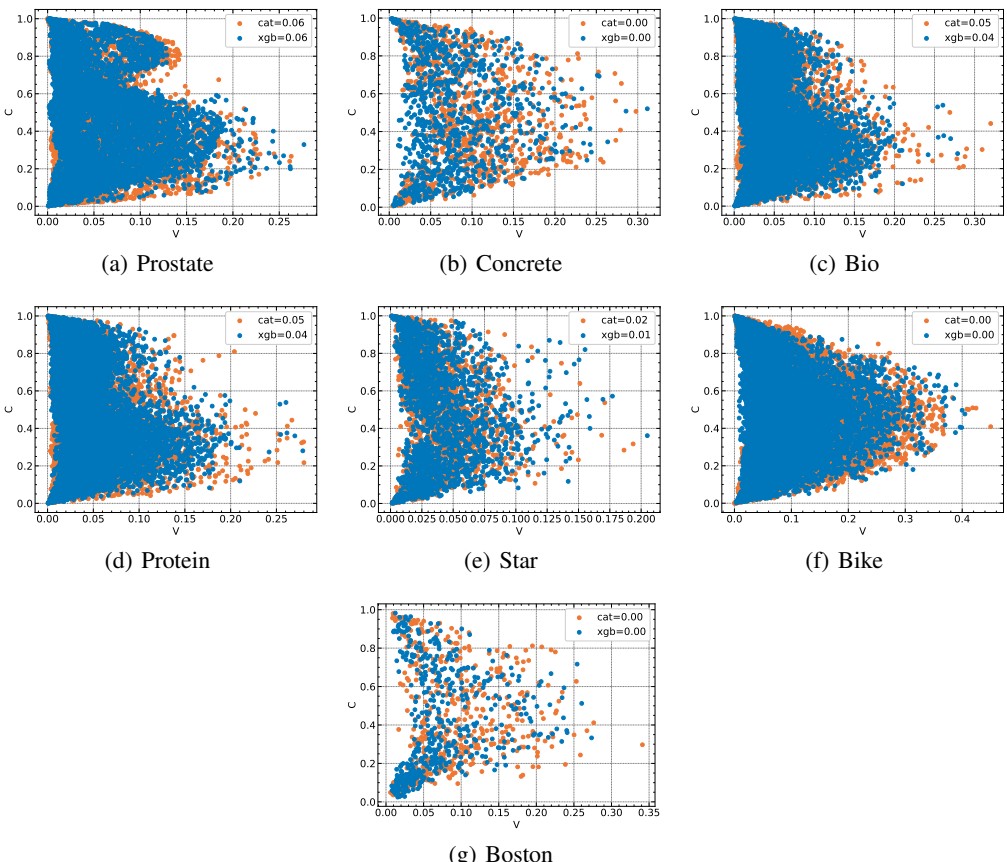

Figure 13: Assessing the consistency & similarity of the TRIAGE characteristic curve using both XGBoost and CatBoost. The characterization by TRIAGE for both methods is similar.

Table 5: Spearman correlation xgboost vs catboost

| Name | Scores corr |
|---|---|
| Bike [35] | 0.77 |
| Bio [35] | 0.94 |
| Boston Housing [35] | 0.8 |
| Concrete [35] | 0.83 |
| Prostate [31] | 0.97 |
| Protein [35] | 0.94 |
| Star [35] | 0.97 |

**PTO - SEE NEXT PAGE FOR CONTINUATION**

## C.4 Fairness - extending sculpting beyond predictive performance

**Goal.** Data sculpting can be used to not only improve predictive performance but also address issues of fairness. For instance, when a dataset contains biases that may harm certain groups at deployment, a smaller, more reflective dataset can guide the sculpting of the larger dataset, improving the fairness of the model trained on it.

**Experiment.** We demonstrate this using a similar calibration approach as in previous experiments, using the Communities and Crime dataset [35, 51], where the regression task is predicting the ViolentCrimesPerPop variable (total number of violent crimes per 100K population) based on a set of demographic variables. Similar to [52], we identify race as a sensitive attribute, often a source of bias. The protected group is considered as those where more than 50% of the community identifies as Black.

We compare TRIAGE based sculpting to a baseline XGBoost (trained without sculpting). We also use the same data-driven and model-driven baselines as the previous experiment. Additionally, we compare to a method that directly optimizes the model for fairness (as opposed to TRIAGE sculpting based on a set of examples) i.e. Fair Regression with Bounded Group Loss [53] [53] — BGL Model.

We evaluate the methods using three fairness metrics for regression from **(author?)** [52]; estimated via Direct Density Ratio Estimation, i.e.
(i) *Independence*: $S \perp A$;
(ii) *Separation*: $S \perp A \mid Y$
(iii) *Sufficiency*: $S \perp A \mid R$.
Where: A - protected group, Y - true target and S - model's prediction. We use the common 80% rule [54, 55] and set $\epsilon = 0.8$ as the threshold for fair assessment, comparing the ratio of metrics between protected and privileged groups, which should fall between $(\epsilon, 1/\epsilon)$, i.e. (0.8, 1.25).

**Results.** We show that TRIAGE-based sculpting has potential beyond predictive tasks, as it can improve fairness metrics without sacrificing performance, as shown in Table 6. Unlike other methods that require direct access to the protected attribute or direct model optimization, TRIAGE uses a set of calibration examples to sculpt the data. We compare TRIAGE to BGL models and Baseline $\mathcal{D}_{\text{cal}}$, which are the only methods to meet the criteria on all three fairness metrics, and show these methods often sacrifice predictive performance to achieve fairness. TRIAGE offers a flexible alternative to improve models for fairness simply based on access to a limited set of examples.

Table 6: Assessment of fairness metrics, highlighting the potential of data sculpting via TRIAGE

| Method | Independence | Separation | Sufficiency | MAE |
|---|---|---|---|---|
| | | *Fairness* | | |
| TRIAGE | 1.21 (✔) | 1.11 (✔) | 1.00 (✔) | 0.0528 |
| Baseline ($\mathcal{D}_{\text{train}}$) | 1.29 (✗) | 1.18 (✔) | 1.01 (✔) | 0.0554 |
| Baseline ($\mathcal{D}_{\text{cal}}$) | 1.14 (✔) | 1.09 (✔) | 0.99 (✔) | 0.0689 |
| Baseline ($\mathcal{D}_{\text{train}} \cup \mathcal{D}_{\text{cal}}$) | 1.27 (✗) | 1.13 (✔) | 1.02 (✔) | 0.0467 |
| Error Sculpt | 1.41 (✗) | 1.19 (✔) | 1.03 (✔) | 0.0502 |
| CP Intervals Sculpt | 1.26 (✗) | 1.15 (✔) | 1.01 (✔) | 0.0467 |
| NGBoost sculpt | 1.47 (✗) | 1.26 (✗) | 1.02 (✔) | 0.0594 |
| Bayesian ridge sculpt | 1.39 (✗) | 1.21 (✔) | 1.00 (✔) | 0.0578 |
| BNN sculpt | 1.27 (✗) | 1.15 (✔) | 1.00 (✔) | 0.0507 |
| GP sculpt | 1.43 (✗) | 1.28 (✗) | 1.02 (✔) | 0.0559 |
| BGL Model 1 | 1.01 (✔) | 1.00 (✔) | 1.02 (✔) | 0.1254 |
| BGL Model 2 | 1.21 (✔) | 1.12 (✔) | 1.00 (✔) | 0.0602 |
| BGL Model 3 | 1.31 (✗) | 1.16 (✔) | 1.0 (✔) | 0.1101 |

**Takeaway.** TRIAGE demonstrates the value of sculpting larger datasets by leveraging a limited set of examples. This emphasizes the importance of fit-for-purpose data and demonstrates how it can not only improve predictive performance but also address important issues such as fairness.

## C.5 Data insights

**Goal.** Besides characterizing examples, it is often useful to try to understand which types of samples are in each group. Specifically, for the fit for purpose experiment, we are trying to understand which samples in $\mathcal{D}_{\text{train}}$ are sculpted as a consequence of $\mathcal{D}_{\text{cal}}$. Recall $\mathcal{D}_{\text{train}}$ are patients from the US and $\mathcal{D}_{\text{cal}}$ and $\mathcal{D}_{\text{test}}$ are patients from the UK.

We visualize the output of TRIAGE based sculpting using a radial plot in Figure 14.

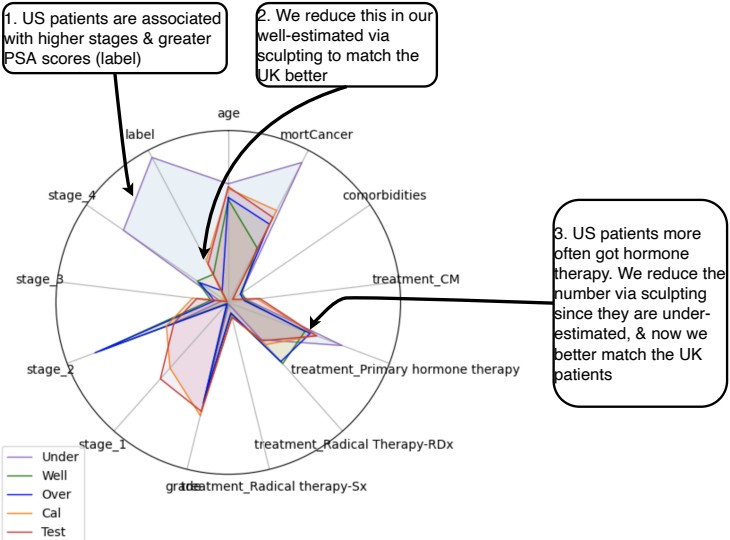

Figure 14: Insights into the different groups resulting from TRIAGE based sculpting. We see we make $\mathcal{D}_{\text{train}}$ look more similar to the UK for many features by sculpting.

**Takeaway.** US patients ($\mathcal{D}_{\text{train}}$) typically have *higher* cancer stages and *higher* PSA scores than their UK counterparts ($\mathcal{D}_{\text{cal}}/\mathcal{D}_{\text{test}}$). It is precisely these uncommon samples that are filtered by TRIAGE to improve predictive performance. i.e. we can see this based on the samples classed as well-estimated. This now makes the sculpted US data look more similar to the UK data, hence improving the performance of a model trained on said data. In addition, the US patients more often get hormone therapy, in the sculpted well-estimated set, we reduce these by filtering to match the UK better.

## C.6 Validity assessment: Calibration curves & CRPS

**Goal.** We also wish to evaluate the CPDs and their quality. Typically, CPDs are valid, meaning they are well-calibrated. There are three typical ways in which Conformal Predictive Distributions (CPDs) are evaluated for such properties like validity and quality.

1. Calibration curves: where the desired curve is diagonal.
2. Computing the Continuous Rank Probability (CRPS) score [56]: measures the quality of the predictive distribution, where CRPS=0 is perfect accuracy, and CRPS=1 is fully inaccurate, given by Equation 5.
3. Assigning quantiles to the CPD and then computing coverage of the resulting intervals. i.e. if we assign quantiles of lower = 0.05 and upper = 0.95, then we desire coverage = 0.9.

$$CRPS(P, x) = \int_{-\infty}^{\infty} ||P(x^*) - H(x^* - x)||_2 dx \tag{5}$$

where $x$ is the true value of $x$, $P(x^*)$ is the proposed predictive distribution, $H(x)$ the Heaviside step function ($H(x) = 1$ if $x = 0$, $H(x) = 0$ if $x \leq 0$).

We evaluate these metrics for two settings:

- **Fine-grained filter**: in this experiment we are IID, hence we satisfy the exchangeability assumption naturally. Consequently, the CPDs have guaranteed validity.
- **Sculpting for a deployment purpose**: in this experiment $\mathcal{D}_{\text{train}}$ and $\mathcal{D}_{\text{cal}}$ might not be exchangeable, hence we wish to assess the potential impact empirically using the different metrics.

### C.6.1 Fine-grained filter

This is the ideal setting since we are IID hence, the data is naturally exchangeable. Hence, we have guarantees about the validity of our CPDs. However, we still empirically assess the CPDs computing the calibration curves for all datasets, shown in Figure 15.

**Takeaway.** We see that all the datasets have calibration curves matching the ideal diagonal line. The CRPS scores are also very low, indicating high-quality predictive distributions. The takeaway is that indeed the CPDs are well-calibrated.

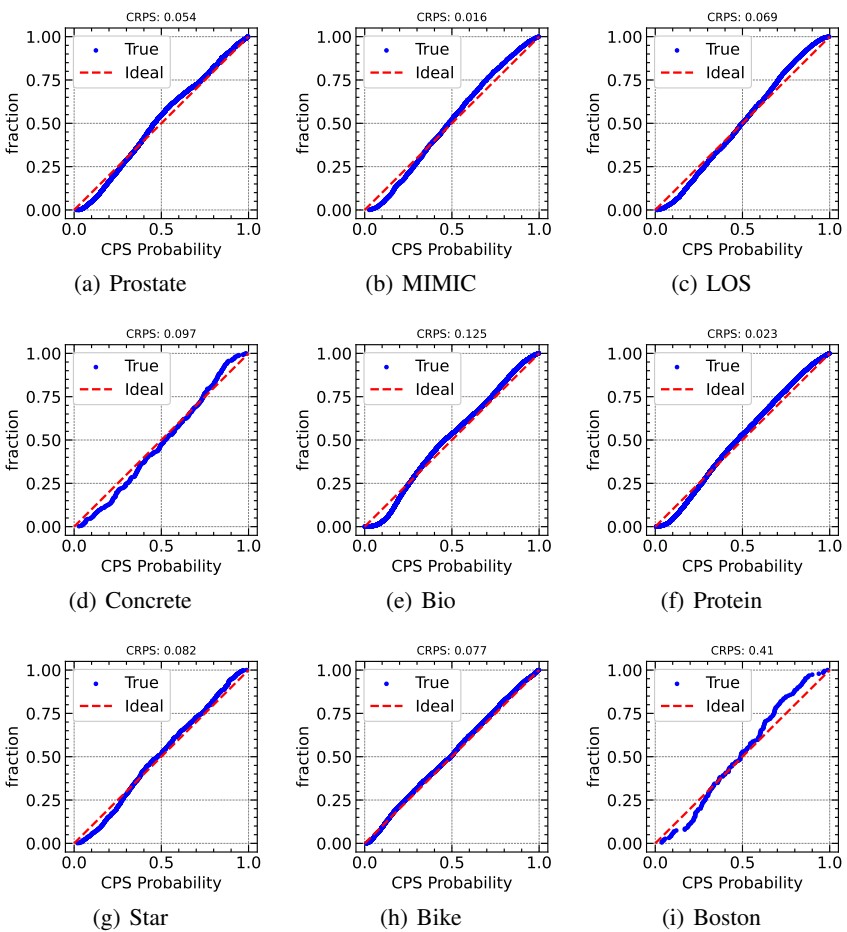

(a) Prostate     (b) MIMIC     (c) LOS

(d) Concrete     (e) Bio     (f) Protein

(g) Star     (h) Bike     (i) Boston

Figure 15: Calibration curves for the different datasets, showing that the CPDs are well calibrated, as they match the ideal

### C.6.2 Sculpting for a deployment purpose

**Goal.** We now assess the sculpting for a deployment purpose case where we sculpt $\mathcal{D}_{\text{train}}$ (US patients) with respect to $\mathcal{D}_{\text{cal}}$ (UK patients). We evaluate this experiment for varying sizes of $\mathcal{D}_{\text{cal}}$ and wish to assess how the potential violation of exchangeability harms calibration, as well as the quality of the CPDs.

**Results.** Figure 16 shows in (a) how coverage varies as the size of $\mathcal{D}_{\text{cal}}$ increases, while (b) looks at how CRPS changes as a function of the size of $\mathcal{D}_{\text{cal}}$. We also check the calibration curves in Figure 17.

**Takeaway.** We see the following

1. When the calibration set $\mathcal{D}_{\text{cal}}$ is small, $< 0.3$ in proportion ($< 200/300$ samples) then we still **achieve marginal coverage** $> 0.9$ — this is also reflected in the low CRPS and good calibration curves.

2. After $0.3$ ($> 300$ samples) we have sufficient samples that violate exchangeability. This reduces the marginal coverage below 0.9. Interestingly, this **matches the change-over point we refer to in the main paper - Table 2**, where training directly on $\mathcal{D}_{\text{cal}}$ will lead to better performance.

We do, however, highlight that the harm is not significant— hence our CPDs are still of high quality, as can be seen by the low CRPS score.

*Consequently, we show empirically, even if this may seem an issue — the resulting CPDs are not significantly harmed by this. Moreover, coverage does not have as dramatic a drop empirically as one might expect*

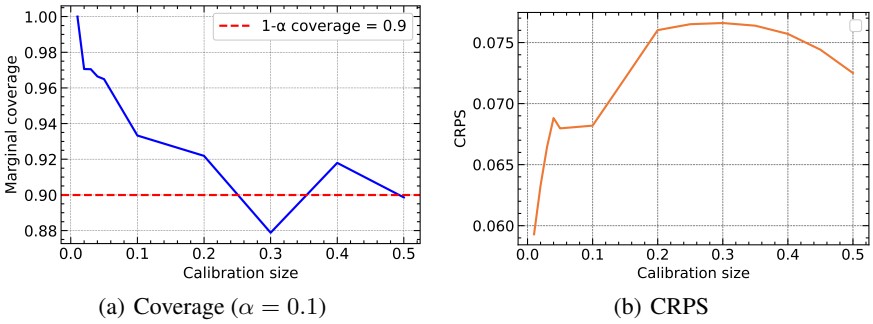

(a) Coverage ($\alpha = 0.1$)  (b) CRPS

Figure 16: Coverage and CRPS as we increase the size of $\mathcal{D}_{\text{cal}}$

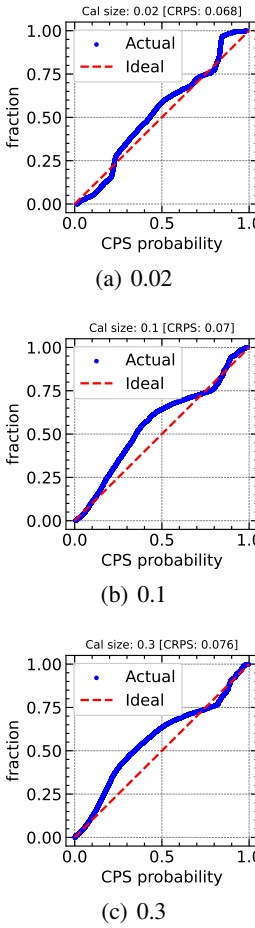

Figure 17: Calibration curves for different sizes of $\mathcal{D}_{\text{cal}}$

## C.7 Sculpting for deployment purpose: imbalance and long tails

In the main text, in Section 5.2.2 we looked at sculpting $\mathcal{D}_{\text{train}}$ guided by $\mathcal{D}_{\text{cal}}$, in order to make the data more "fit for purpose". In this setup, we looked at the data on average. We now take a deeper dive, looking at: (i) Data imbalance (Majority and minority groups) and (ii) Long tails of outcomes

### C.7.1 Data Imbalance

**Goal.** The dataset is heavily skewed towards older patients $> 65$ years old, with very few younger patients by virtue of the nature of prostate cancer. We wish to understand, beyond average, what is the impact of sculpting on performance of different subsets. In particular, the majority (older) and minority (younger) groups.

We report the results in Table 7 and see the following results. Note: TRIAGE (REST) refers to over- and under-estimated samples used, whilst TRIAGE (OURS) refers to only using well-estimated samples:

**Takeaway.** In the case of data imbalance, TRIAGE demonstrates the utility of sculpting the data to be more fit for purpose — especially in the small sample regime. After around 200 examples, there is reduced benefit to sculpting a pre-existing larger dataset to be more fit for purpose, but rather to directly train on the given examples. We see this benefit also under data imbalance (via attributes) across both majority and minority groups.

Overall, the behavior we see here is similar to the main experiment on average behavior.

Table 7: Data Imbalance: Comparison of MSE for different approaches as we change the number of prior examples we have can access.

| $\mathcal{D}_{\text{cal}}$ examples | 10 | 20 | 30 | 40 | 50 | 100 | 200 | 300 | 400 | 500 |
|---|---|---|---|---|---|---|---|---|---|---|
| **Overall - all samples** | | | | | | | | | | |
| TRIAGE (Ours) - $WE$ | **0.038+-0.002** | **0.037+-0.002** | **0.038+-0.002** | **0.04+-0.002** | **0.04+-0.002** | **0.041+-0.002** | 0.039+-0.002 | 0.039+-0.001 | 0.039+-0.001 | 0.04+-0.001 |
| TRIAGE (REST) $OE \cup UE$ | 0.104+-0.019 | 0.104+-0.018 | 0.107+-0.023 | 0.109+-0.016 | 0.101+-0.019 | 0.124+-0.021 | 0.113+-0.023 | 0.109+-0.021 | 0.115+-0.02 | 0.111+-0.02 |
| Baseline ($\mathcal{D}_{\text{train}}$) | 0.085+-0.011 | 0.085+-0.011 | 0.085+-0.011 | 0.085+-0.011 | 0.085+-0.011 | 0.085+-0.011 | 0.085+-0.011 | 0.085+-0.011 | 0.085+-0.011 | 0.085+-0.011 |
| Baseline ($\mathcal{D}_{\text{cal}}$) | 0.05+-0.01 | 0.047+-0.011 | 0.045+-0.004 | 0.040+-0.003 | 0.041+-0.002 | 0.041+-0.003 | **0.034+-0.001** | **0.033+-0.001** | **0.033+-0.001** | 0.032+-0.001 |
| Baseline ($\mathcal{D}_{\text{train}} \cup \mathcal{D}_{\text{cal}}$) | 0.071+-0.01 | 0.062+-0.006 | 0.059+-0.006 | 0.054+-0.005 | 0.051+-0.005 | 0.044+-0.004 | 0.039+-0.003 | 0.035+-0.002 | **0.033+-0.002** | **0.031+-0.001** |
| Error Sculpt | 0.078+-0.02 | 0.077+-0.019 | 0.077+-0.019 | 0.062+-0.019 | 0.076+-0.019 | 0.078+-0.018 | 0.076+-0.019 | 0.073+-0.019 | 0.076+-0.018 | 0.077+-0.019 |
| CP Intervals Sculpt | 0.09+-0.016 | 0.061+-0.022 | 0.045+-0.004 | 0.059+-0.015 | 0.052+-0.006 | 0.054+-0.013 | 0.059+-0.02 | 0.037+-0.001 | 0.034+-0.001 | 0.039+-0.002 |
| NGBoost | 0.076+-0.012 | 0.086+-0.011 | 0.083+-0.021 | 0.085+-0.01 | 0.102+-0.017 | 0.109+-0.038 | 0.089+-0.018 | 0.096+-0.015 | 0.084+-0.011 | 0.097+-0.019 |
| Bayesian ridge | 0.092+-0.015 | 0.102+-0.023 | 0.098+-0.018 | 0.103+-0.018 | 0.117+-0.025 | 0.094+-0.017 | 0.102+-0.015 | 0.114+-0.028 | 0.115+-0.034 | 0.099+-0.02 |
| BNN | 0.081+-0.008 | 0.088+-0.013 | 0.062+-0.007 | 0.101+-0.012 | 0.095+-0.017 | 0.11+-0.015 | 0.092+-0.023 | 0.078+-0.011 | 0.078+-0.009 | 0.077+-0.01 |
| GP | 0.122+-0.023 | 0.117+-0.016 | 0.12+-0.019 | 0.113+-0.018 | 0.108+-0.013 | 0.125+-0.017 | 0.131+-0.016 | 0.13+-0.021 | 0.145+-0.024 | 0.144+-0.024 |
| $\mathcal{D}_{\text{cal}}$ examples | 10 | 20 | 30 | 40 | 50 | 100 | 200 | 300 | 400 | 500 |
| **Majority group (75%)** | | | | | | | | | | |
| TRIAGE (Ours) - $WE$ | **0.042+-0.004** | **0.041+-0.002** | **0.042+-0.003** | **0.042+-0.002** | **0.044+-0.003** | 0.046+-0.002 | 0.043+-0.003 | 0.043+-0.002 | 0.044+-0.002 | 0.045+-0.001 |
| TRIAGE (Rest) - $OE \cup UE$ | 0.09+-0.018 | 0.093+-0.018 | 0.095+-0.024 | 0.099+-0.018 | 0.086+-0.018 | 0.106+-0.022 | 0.101+-0.023 | 0.093+-0.021 | 0.099+-0.021 | 0.097+-0.02 |
| Baseline ($\mathcal{D}_{\text{train}}$) | 0.075+-0.013 | 0.075+-0.013 | 0.075+-0.013 | 0.075+-0.013 | 0.075+-0.013 | 0.075+-0.013 | 0.075+-0.013 | 0.075+-0.013 | 0.075+-0.013 | 0.075+-0.013 |
| Baseline ($\mathcal{D}_{\text{cal}}$) | 0.055+-0.012 | 0.054+-0.014 | 0.052+-0.004 | 0.044+-0.003 | 0.047+-0.002 | 0.045+-0.003 | 0.037+-0.001 | 0.036+-0.001 | 0.036+-0.001 | 0.035+-0.001 |
| Baseline ($\mathcal{D}_{\text{train}} \cup \mathcal{D}_{\text{cal}}$) | 0.056+-0.012 | 0.048+-0.006 | 0.045+-0.005 | 0.042+-0.003 | **0.044+-0.003** | **0.037+-0.002** | **0.034+-0.001** | **0.032+-0.001** | **0.032+-0.001** | **0.031+-0.001** |
| Error Sculpt | 0.038+-0.002 | 0.039+-0.002 | 0.039+-0.002 | 0.039+-0.002 | 0.039+-0.002 | 0.041+-0.004 | 0.038+-0.002 | 0.038+-0.002 | 0.038+-0.002 | 0.038+-0.002 |
| CP Intervals Sculpt | 0.073+-0.012 | 0.06+-0.022 | 0.044+-0.005 | 0.055+-0.013 | 0.053+-0.004 | 0.056+-0.017 | 0.046+-0.005 | 0.039+-0.002 | 0.037+-0.001 | 0.042+-0.002 |
| NGBoost | 0.06+-0.01 | 0.073+-0.01 | 0.071+-0.02 | 0.071+-0.009 | 0.097+-0.021 | 0.098+-0.039 | 0.077+-0.02 | 0.08+-0.012 | 0.073+-0.012 | 0.086+-0.02 |
| Bayesian ridge | 0.083+-0.017 | 0.099+-0.027 | 0.088+-0.022 | 0.096+-0.02 | 0.114+-0.032 | 0.085+-0.018 | 0.092+-0.014 | 0.109+-0.036 | 0.111+-0.042 | 0.092+-0.024 |
| BNN | 0.074+-0.009 | 0.075+-0.018 | 0.05+-0.006 | 0.086+-0.012 | 0.086+-0.019 | 0.106+-0.019 | 0.081+-0.025 | 0.067+-0.009 | 0.066+-0.008 | 0.062+-0.007 |
| GP | 0.102+-0.025 | 0.094+-0.014 | 0.095+-0.012 | 0.098+-0.016 | 0.094+-0.009 | 0.105+-0.021 | 0.111+-0.019 | 0.113+-0.02 | 0.129+-0.024 | 0.132+-0.026 |
| $\mathcal{D}_{\text{cal}}$ examples | 10 | 20 | 30 | 40 | 50 | 100 | 200 | 300 | 400 | 500 |
| **Minority group (25%)** | | | | | | | | | | |
| TRIAGE (Ours) - $WE$ | **0.028+-0.004** | **0.026+-0.002** | **0.026+-0.002** | **0.027+-0.003** | **0.026+-0.002** | **0.026+-0.002** | **0.026+-0.002** | 0.026+-0.003 | 0.026+-0.002 | 0.027+-0.003 |
| TRIAGE (Rest) - $OE \cup UE$ | 0.143+-0.027 | 0.135+-0.021 | 0.14+-0.023 | 0.138+-0.014 | 0.141+-0.026 | 0.175+-0.022 | 0.149+-0.027 | 0.155+-0.024 | 0.161+-0.024 | 0.153+-0.027 |
| Baseline ($\mathcal{D}_{\text{train}}$) | 0.115+-0.016 | 0.115+-0.016 | 0.115+-0.016 | 0.115+-0.016 | 0.115+-0.016 | 0.115+-0.016 | 0.115+-0.016 | 0.115+-0.016 | 0.115+-0.016 | 0.115+-0.016 |
| Baseline ($\mathcal{D}_{\text{cal}}$) | 0.036+-0.008 | 0.027+-0.003 | 0.027+-0.004 | 0.025+-0.002 | 0.026+-0.003 | 0.028+-0.005 | **0.026+-0.001** | **0.025+-0.001** | **0.024+-0.001** | **0.024+-0.002** |
| Baseline ($\mathcal{D}_{\text{train}} \cup \mathcal{D}_{\text{cal}}$) | 0.114+-0.015 | 0.103+-0.015 | 0.1+-0.017 | 0.0889+-0.017 | 0.082+-0.015 | 0.066+-0.015 | 0.054+-0.009 | 0.044+-0.007 | 0.037+-0.004 | 0.033+-0.003 |
| Error Sculpt | 0.192+-0.074 | 0.185+-0.073 | 0.187+-0.073 | 0.131+-0.071 | 0.18+-0.071 | 0.183+-0.07 | 0.184+-0.071 | 0.173+-0.071 | 0.182+-0.069 | 0.189+-0.071 |
| CP Intervals Sculpt | 0.138+-0.034 | 0.063+-0.023 | 0.046+-0.009 | 0.07+-0.022 | 0.048+-0.01 | 0.047+-0.017 | 0.098+-0.069 | 0.029+-0.001 | 0.026+-0.002 | 0.028+-0.003 |
| NGBoost | 0.122+-0.027 | 0.124+-0.023 | 0.117+-0.029 | 0.123+-0.018 | 0.115+-0.013 | 0.142+-0.037 | 0.121+-0.016 | 0.142+-0.026 | 0.116+-0.02 | 0.128+-0.022 |
| Bayesian ridge | 0.114+-0.017 | 0.11+-0.015 | 0.123+-0.016 | 0.122+-0.017 | 0.124+-0.016 | 0.122+-0.022 | 0.129+-0.022 | 0.127+-0.022 | 0.125+-0.021 | 0.119+-0.02 |
| BNN | 0.103+-0.014 | 0.123+-0.023 | 0.095+-0.014 | 0.147+-0.023 | 0.118+-0.016 | 0.122+-0.022 | 0.122+-0.021 | 0.108+-0.022 | 0.109+-0.017 | 0.118+-0.029 |
| GP | 0.18+-0.033 | 0.183+-0.034 | 0.193+-0.043 | 0.154+-0.034 | 0.149+-0.035 | 0.182+-0.023 | 0.191+-0.023 | 0.18+-0.035 | 0.19+-0.036 | 0.179+-0.031 |

We look into the issue of long tail next

### C.7.2 Long tail of outcomes

**Goal.** Going beyond imbalance on the feature-level, there are also long-tails of outcomes. In this case, the long tails are few samples are associated with high prostate cancer scores. So we partition based on the outcome where the tail greater than 0.75 normed score represents only 3% of the data, whilst the remainder is 97%.

We seek to understand the performance via sculpting on both the head and tail of the distributions to assses the impact.

The results in Table 8 show the following results. Note: TRIAGE (REST) refers to over- and under-estimated samples used, whilst TRIAGE (OURS) refers to only using well-estimated samples:

**Takeaway.** In the case of having long tails on the outcome, TRIAGE demonstrates the utility of sculpting the data to be more fit for purpose, when assessed overall (averaged across all samples) — especially in the small sample regime. After around 200 examples, there is reduced benefit to sculpting a pre-existing larger dataset to be more fit for purpose, but rather to directly train on the given examples.

Head: For the head of the distribution which represents the majority of the data this is also the case and we see the best performance for TRIAGE ($WE$).

Tails: On the long tails of the outcome distribution, we see that in fact using the TRIAGE (REST) - consisting of the "filtered" examples ($OE \cup UE$) has the best performance. This result highlights an alternative example selection enabled by TRIAGE. i.e. we should train a specific and better performing model for the long tails specifically using $OE \cup UE$.

Table 8: Long tail: Comparison of MSE for different approaches as we change the number of prior examples we have can access.

**Overall all samples**

| $\mathcal{D}_{cal}$ examples | 10 | 20 | 30 | 40 | 50 | 100 | 200 | 300 | 400 | 500 |
|---|---|---|---|---|---|---|---|---|---|---|
| TRIAGE (Ours) - $WE$ | **0.04+-0.002** | **0.037+-0.002** | **0.037+-0.002** | **0.04+-0.002** | **0.04+-0.002** | **0.04+-0.002** | 0.039+-0.002 | 0.039+-0.001 | 0.039+-0.001 | 0.04+-0.0 |
| TRIAGE (Rest) - $OE \cup UE$ | 0.103+-0.02 | 0.102+-0.019 | 0.104+-0.017 | 0.104+-0.018 | 0.105+-0.022 | 0.125+-0.022 | 0.114+-0.023 | 0.112+-0.022 | 0.115+-0.021 | 0.112+-0.021 |
| Baseline ($\mathcal{D}_{train}$) | 0.085+-0.011 | 0.085+-0.011 | 0.085+-0.011 | 0.085+-0.011 | 0.085+-0.011 | 0.085+-0.011 | 0.085+-0.011 | 0.085+-0.011 | 0.085+-0.011 | 0.085+-0.011 |
| Baseline ($\mathcal{D}_{cal}$) | 0.05+-0.01 | 0.047+-0.011 | 0.045+-0.004 | 0.039+-0.003 | 0.041+-0.002 | 0.041+-0.003 | **0.034+-0.001** | **0.033+-0.001** | **0.033+-0.001** | 0.032+-0.001 |
| Baseline ($\mathcal{D}_{train} \cup \mathcal{D}_{cal}$) | 0.071+-0.01 | 0.062+-0.006 | 0.059+-0.006 | 0.054+-0.005 | 0.051+-0.005 | 0.044+-0.004 | 0.039+-0.003 | 0.035+-0.002 | **0.033+-0.002** | **0.031+-0.001** |
| Error Sculpt | 0.08+-0.019 | 0.078+-0.019 | 0.078+-0.019 | 0.062+-0.019 | 0.075+-0.019 | 0.078+-0.018 | 0.077+-0.019 | 0.073+-0.018 | 0.076+-0.018 | 0.077+-0.019 |
| CP Intervals Sculpt | 0.091+-0.017 | 0.063+-0.025 | 0.044+-0.004 | 0.058+-0.015 | 0.05+-0.006 | 0.053+-0.012 | 0.058+-0.019 | 0.036+-0.001 | 0.034+-0.001 | 0.039+-0.002 |
| NGBoost | 0.075+-0.011 | 0.082+-0.009 | 0.082+-0.022 | 0.087+-0.009 | 0.106+-0.021 | 0.109+-0.038 | 0.089+-0.018 | 0.098+-0.016 | 0.084+-0.011 | 0.098+-0.019 |
| Bayesian ridge | 0.091+-0.015 | 0.093+-0.023 | 0.097+-0.018 | 0.104+-0.018 | 0.119+-0.026 | 0.094+-0.017 | 0.104+-0.013 | 0.112+-0.029 | 0.113+-0.032 | 0.099+-0.02 |
| BNN | 0.091+-0.015 | 0.114+-0.027 | 0.099+-0.025 | 0.082+-0.014 | 0.075+-0.01 | 0.072+-0.006 | 0.079+-0.014 | 0.065+-0.009 | 0.073+-0.009 | 0.095+-0.018 |
| GP | 0.111+-0.024 | 0.117+-0.018 | 0.12+-0.018 | 0.113+-0.018 | 0.108+-0.013 | 0.125+-0.016 | 0.135+-0.014 | 0.129+-0.021 | 0.146+-0.024 | 0.142+-0.024 |

**Tail of distribution (3%)**

| $\mathcal{D}_{cal}$ examples | 10 | 20 | 30 | 40 | 50 | 100 | 200 | 300 | 400 | 500 |
|---|---|---|---|---|---|---|---|---|---|---|
| TRIAGE (Ours) - $WE$ | 0.574+-0.028 | 0.57+-0.019 | 0.582+-0.025 | 0.604+-0.026 | 0.61+-0.019 | 0.616+-0.016 | 0.597+-0.023 | 0.603+-0.02 | 0.611+-0.018 | 0.621+-0.016 |
| TRIAGE (Rest) - $OE \cup UE$ | **0.272+-0.017** | **0.289+-0.036** | **0.264+-0.025** | **0.256+-0.032** | **0.224+-0.021** | **0.25+-0.035** | **0.246+-0.033** | **0.268+-0.028** | **0.262+-0.037** | **0.268+-0.038** |
| Baseline ($\mathcal{D}_{train}$) | 0.315+-0.029 | 0.315+-0.029 | 0.315+-0.029 | 0.315+-0.029 | 0.315+-0.029 | 0.315+-0.029 | 0.315+-0.029 | 0.315+-0.029 | 0.315+-0.029 | 0.315+-0.029 |
| Baseline ($\mathcal{D}_{cal}$) | 0.475+-0.05 | 0.456+-0.042 | 0.424+-0.047 | 0.426+-0.033 | 0.399+-0.026 | 0.382+-0.018 | 0.374+-0.014 | 0.386+-0.024 | 0.39+-0.02 | 0.378+-0.023 |
| Baseline ($\mathcal{D}_{train} \cup \mathcal{D}_{cal}$) | 0.366+-0.032 | 0.364+-0.024 | 0.377+-0.035 | 0.388+-0.028 | 0.389+-0.026 | 0.395+-0.025 | 0.408+-0.018 | 0.407+-0.016 | 0.402+-0.017 | 0.398+-0.017 |
| Error Sculpt | 0.503+-0.022 | 0.504+-0.021 | 0.504+-0.024 | 0.502+-0.020 | 0.501+-0.02 | 0.492+-0.026 | 0.488+-0.027 | 0.494+-0.028 | 0.498+-0.028 | 0.495+-0.026 |
| CP Intervals Sculpt | 0.346+-0.03 | 0.433+-0.019 | 0.445+-0.050 | 0.447+-0.065 | 0.39+-0.052 | 0.512+-0.033 | 0.526+-0.022 | 0.532+-0.020 | 0.504+-0.025 | 0.509+-0.029 |
| NGBoost | 0.42+-0.04 | 0.369+-0.028 | 0.379+-0.019 | 0.324+-0.018 | 0.293+-0.018 | 0.402+-0.025 | 0.425+-0.062 | 0.359+-0.024 | 0.414+-0.049 | 0.408+-0.052 |
| Bayesian ridge | 0.357+-0.025 | 0.341+-0.042 | 0.332+-0.045 | 0.333+-0.050 | 0.371+-0.053 | 0.339+-0.036 | 0.299+-0.027 | 0.339+-0.047 | 0.34+-0.039 | 0.341+-0.053 |
| BNN | 0.311+-0.032 | 0.367+-0.046 | 0.327+-0.043 | 0.373+-0.034 | 0.364+-0.053 | 0.368+-0.027 | 0.413+-0.054 | 0.370+-0.026 | 0.377+-0.040 | 0.348+-0.060 |
| GP | 0.359+-0.055 | 0.302+-0.036 | 0.303+-0.033 | 0.321+-0.032 | 0.32+-0.022 | 0.333+-0.034 | 0.337+-0.049 | 0.357+-0.057 | 0.331+-0.045 | 0.328+-0.042 |

**Head of distribution (97%)**

| $\mathcal{D}_{cal}$ examples | 10 | 20 | 30 | 40 | 50 | 100 | 200 | 300 | 400 | 500 |
|---|---|---|---|---|---|---|---|---|---|---|
| TRIAGE (Ours) - $WE$ | **0.028+-0.001** | **0.025+-0.002** | **0.025+-0.002** | **0.027+-0.001** | **0.027+-0.002** | **0.027+-0.002** | **0.026+-0.002** | 0.026+-0.002 | 0.026+-0.001 | 0.027+-0.001 |
| TRIAGE (Rest)- $OE \cup UE$ | 0.099+-0.02 | 0.098+-0.019 | 0.101+-0.018 | 0.101+-0.018 | 0.103+-0.022 | 0.122+-0.023 | 0.111+-0.024 | 0.109+-0.022 | 0.112+-0.022 | 0.109+-0.022 |
| Baseline ($\mathcal{D}_{train}$) | 0.080+-0.012 | 0.080+-0.012 | 0.080+-0.012 | 0.080+-0.012 | 0.080+-0.012 | 0.080+-0.012 | 0.080+-0.012 | 0.080+-0.012 | 0.080+-0.012 | 0.080+-0.012 |
| Baseline ($\mathcal{D}_{cal}$) | 0.040+-0.011 | 0.037+-0.012 | 0.037+-0.004 | 0.030+-0.003 | 0.033+-0.002 | 0.033+-0.002 | **0.026+-0.001** | **0.025+-0.001** | **0.024+-0.001** | 0.024+-0.001 |
| Baseline ($\mathcal{D}_{train} \cup \mathcal{D}_{cal}$) | 0.065+-0.011 | 0.056+-0.006 | 0.052+-0.006 | 0.047+-0.005 | 0.043+-0.005 | 0.036+-0.004 | 0.031+-0.003 | 0.027+-0.002 | 0.025+-0.001 | **0.023+-0.001** |
| Error Sculpt | 0.071+-0.019 | 0.069+-0.020 | 0.068+-0.020 | 0.068+-0.020 | 0.065+-0.019 | 0.068+-0.018 | 0.067+-0.019 | 0.064+-0.019 | 0.066+-0.019 | 0.068+-0.019 |
| CP Intervals Sculpt | 0.085+-0.017 | 0.054+-0.026 | 0.035+-0.005 | 0.049+-0.016 | 0.043+-0.007 | 0.043+-0.013 | 0.048+-0.02 | 0.025+-0.002 | 0.023+-0.001 | 0.028+-0.002 |
| NGBoost | 0.067+-0.012 | 0.076+-0.009 | 0.076+-0.023 | 0.082+-0.008 | 0.102+-0.021 | 0.102+-0.039 | 0.081+-0.019 | 0.093+-0.016 | 0.076+-0.013 | 0.091+-0.02 |
| Bayesian ridge | 0.085+-0.015 | 0.087+-0.024 | 0.092+-0.018 | 0.099+-0.018 | 0.113+-0.027 | 0.088+-0.018 | 0.099+-0.013 | 0.108+-0.030 | 0.108+-0.033 | 0.094+-0.021 |
| BNN | 0.086+-0.015 | 0.108+-0.029 | 0.094+-0.026 | 0.076+-0.015 | 0.068+-0.010 | 0.066+-0.006 | 0.072+-0.016 | 0.058+-0.009 | 0.066+-0.010 | 0.089+-0.019 |
| GP | 0.105+-0.025 | 0.113+-0.019 | 0.116+-0.019 | 0.109+-0.018 | 0.103+-0.013 | 0.120+-0.017 | 0.131+-0.015 | 0.124+-0.022 | 0.142+-0.025 | 0.138+-0.025 |

## C.8 Comparison with Data Valuation

**Goal.** We have presented a conceptual difference of TRIAGE to data valuation methods in Section 2. We now experimentally compare TRIAGE with using data valuation methods for data selection. Specifically, we compare to two Shapley valuation methods: (i) TMC-Shapley [22], (ii) KNN-Shapley [23] and LAVA [24], which uses optimal transport. We compare in the context of Table 2- data selection for sculpting. The TRIAGE $\mathcal{D}_{cal}$ is used as the validation set for these methods.

Note: TMC-Shapley: was computationally unfeasible for all 20k samples. Hence, we sample 5k and compare it to TRIAGE separately. (ii) KNN-Shapley and LAVA is run on the original 20k samples.

**Takeaway.** *Performance:* The results are shown in Tables 9 and 10. While, these methods are competitive with TRIAGE, we find that TRIAGE tailored to regression outperforms them on downstream MSE performance.

*Computational time:* We assess their compute time vs TRIAGE. In Figure 18, we show for different sizes of $\mathcal{D}_{cal}$ that TRIAGE is more time efficient. Additionally, unlike Shapley methods, our cost doesn't increase with the size of $\mathcal{D}_{cal}$. KNN-Shapley is 1-3X and TMC-Shapley 600X more expensive than TRIAGE.

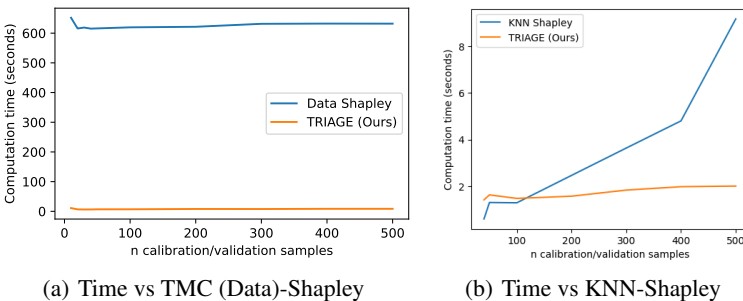

(a) Time vs TMC (Data)-Shapley      (b) Time vs KNN-Shapley

Figure 18: Computational Time (in seconds): (a) TRIAGE is almost 600X more time efficient than TMC-Shapley, (b) TRIAGE is more efficient than KNN-Shapley, especially as $\mathcal{D}_{cal}$'s size increases.

Table 9: Comparison of TRIAGE to Data valuation for different sizes of $\mathcal{D}_{cal}$. TRIAGE's approach to data sculpting outperforms with lower test MSE ($\downarrow$ better). These rows will be added to Table 2 (main paper)

|  | $\mathcal{D}_{cal}$ sample size | 10 | 20 | 30 | 40 | 50 | 100 | 200 | 300 |
|---|---|---|---|---|---|---|---|---|---|
| Ours (Sculpting) | **TRIAGE** ($WE$) | **0.051** | **0.050** | **0.046** | **0.046** | **0.046** | **0.046** | **0.045** | **0.045** |
| Data Valuation | **KNN-Shapley** | 0.092 | 0.095 | 0.092 | 0.122 | 0.115 | 0.086 | 0.054 | 0.075 |
|  | **LAVA** | 0.055 | 0.054 | 0.058 | 0.055 | 0.054 | 0.055 | 0.058 | 0.056 |

Table 10: Comparison of TRIAGE to TMC-Shapley for data sculpting for different sizes of $\mathcal{D}_{cal}$. Due to computational infeasability of TMC-Shapley we subsample to 5k and compare to TRIAGE. TRIAGE's approach to sculpting has lower test MSE mostly compared to TMC-Shapley ($\downarrow$ better).

|  | $\mathcal{D}_{cal}$ sample size | 10 | 20 | 30 | 40 | 50 | 100 | 200 | 300 |
|---|---|---|---|---|---|---|---|---|---|
| Ours (Sculpting) | **TRIAGE** ($WE$) | **0.0477** | 0.0488 | 0.0498 | **0.0457** | **0.0447** | **0.0468** | **0.0470** | **0.0461** |
| Data Valuation | **TMC-Shapley** | 0.0580 | **0.0441** | **0.0490** | 0.0472 | 0.0580 | 0.0491 | 0.0501 | 0.0467 |

## C.9 Synthetic analysis under Huber's contamination model

TRIAGE aims to sculpt data for robust regression. One can of course draw similarities to robust statistics which also aims to train robust models. (1) Post-hoc vs Built-in: TRIAGE wraps a regressor to detect and sculpt outlier samples, whereas Robust Statistics embeds outlier resilience within the model [57, 58], e.g. via Huber loss [58]. (2) Additional data-centric applications: TRIAGE tackles diverse "data-centric AI" tasks, like comparing synthetic data (Sec. 5.3.1) and feature acquisition/collection (Sec. 5.3.2), which is beyond the scope of robust statistics.

(ii) Theoretical Analysis: Connecting TRIAGE theoretically to robust statistics is an intriguing question. However, we highlight two important challenges that could be tackled by future work, for inspiration see [57]:

(1) Interdependence of CPD and training dynamics in TRIAGE. Disentangling their impacts theoretically is non-trivial. (2) Scores across training epochs are correlated due to iterative model training. This highlights the complexity of any theoretical guarantee, given the dynamic nature of the scores and their correlation.

As a step towards this we contrast TRIAGE with robust statistics. We provide a simulation using Huber's Contamination Model: Our simulation setup mirrors [59], generating data from a linear model $y = X\beta + \eta$, where $X \sim U[0, 1]$ and $\eta \sim N(0, 1)$.

Mimicking Huber's model we contaminate the response $y$ corrupting $\epsilon$ samples replacing $\eta_i$ with $\eta_i + c_i$, where $c_i$ comes from a different distribution.

We compare TRIAGE with: (i) Error baseline, (ii) Training with Huber Loss, (iii) TRIAGE applied to a Huber Loss trained model.

The results in Figure 19 show that TRIAGE has a lower MSE compared to the error baseline as $\epsilon$ rises. TRIAGE is also stable in response to contamination due to the clean calibration set. Interestingly, combining TRIAGE with a model trained using Huber's loss proves superior to using either alone, highlighting the compatibility of TRIAGE with robust techniques.

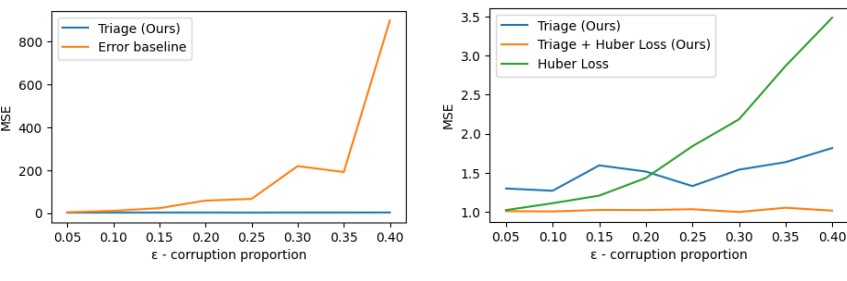

(a) Comparison to error baseline      (b) TRIAGE is compatible with robust losses

Figure 19: Simulation with Huber's contamination model: (a) TRIAGE has lower MSE as $\epsilon$ increases and (b) TRIAGE combined with a model trained with a Huber loss has lower MSE than both alone.

## C.10 Differentiation of general-purpose scoring methods with TRIAGE scores

**Goal.** The main text (Section 5.2) illustrated that for the same magnitude of general-purpose scores these can be associated with different TRIAGE scores. Hence, highlighting that TRIAGE scores offer a viable alternative to differentiating samples.

In the main text, we looked at error vs TRIAGE scores . We now examine the remainder of general purpose scoring methods in Figure 20.

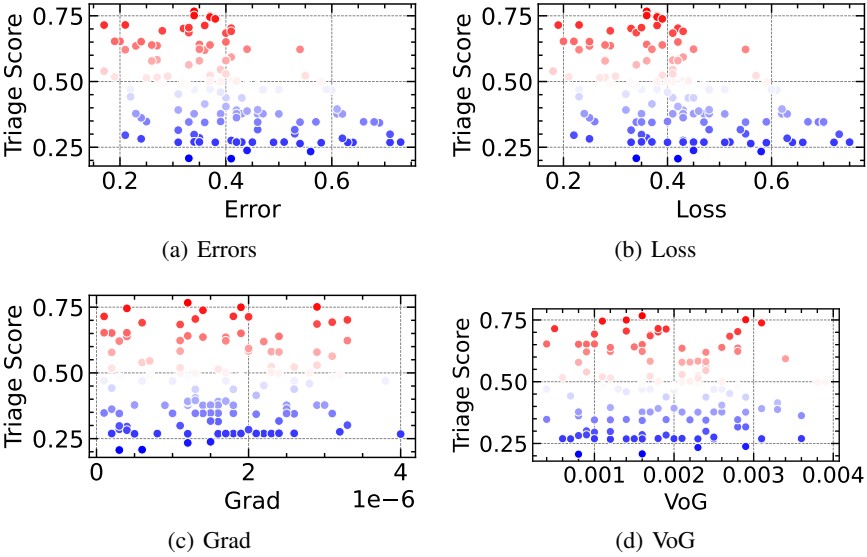

Figure 20: Samples with the same general-purpose scores are often associated with, different Triage scores, highlighting the potential to differentiate sample

**Takeaway.** The phenomena where TRIAGE scores can be used to differentiate samples with the same magnitude holds true across the different scoring approaches.

## C.11 Fine-grained filtering: Additional datasets

**Goal.** In the main paper we demonstrated fine-grained filtering on a dataset to showcase the potential. We now repeat on all datasets.

**Takeaway.** The results shown in Figures 21-28 highlight that indeed less is more. Fitting on more high-quality samples can result it better performance.

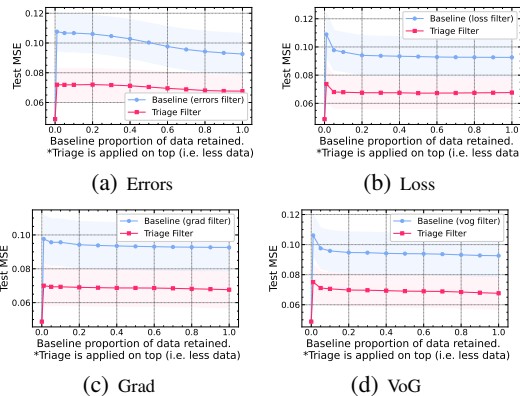

Figure 21: Fine-grained filter: Bike

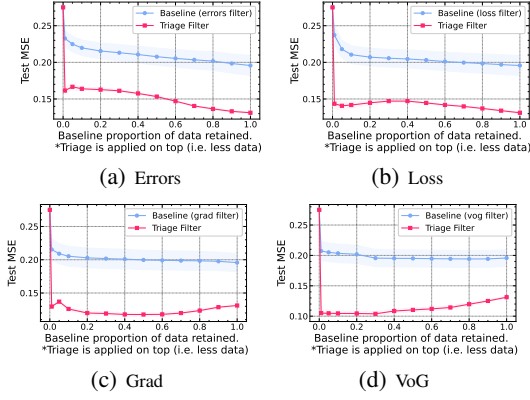

Figure 22: Fine-grained filter: Bio

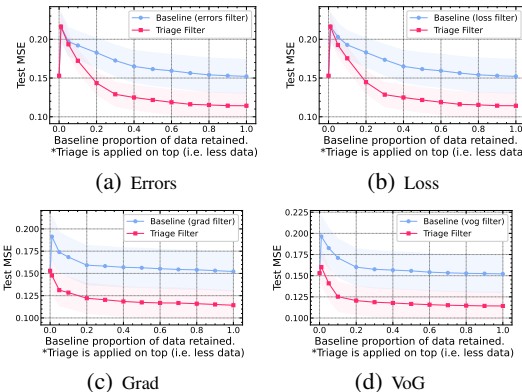

Figure 23: Fine-grained filter: Concrete

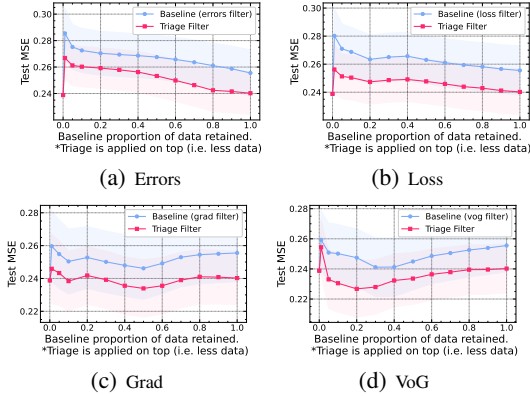

(a) Errors           (b) Loss

(c) Grad           (d) VoG

Figure 24: Fine-grained filter: Prostate

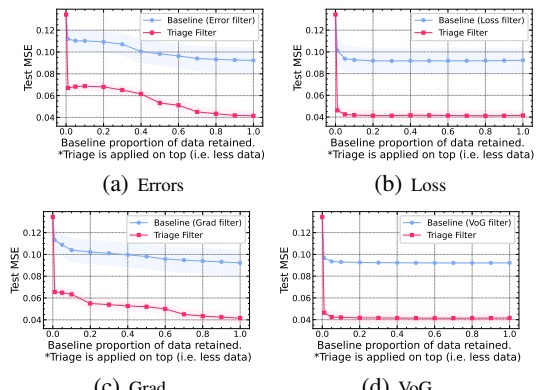

(a) Errors           (b) Loss

(c) Grad           (d) VoG

Figure 25: Fine-grained filter: LoS

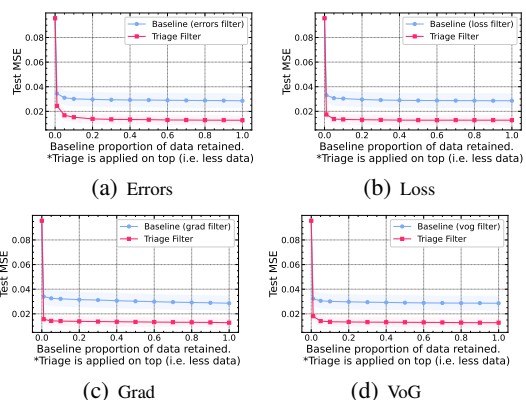

(a) Errors           (b) Loss

(c) Grad           (d) VoG

Figure 26: Fine-grained filter: Mimic

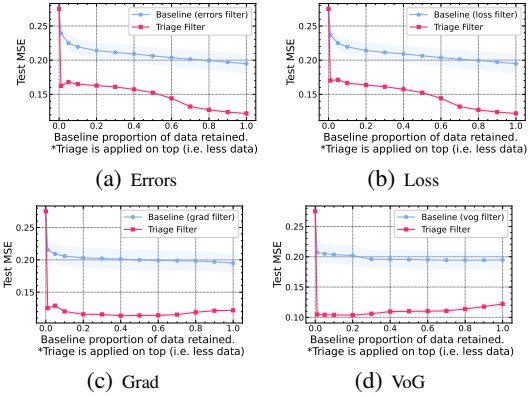

(a) Errors

(b) Loss

(c) Grad

(d) VoG

Figure 27: Fine-grained filter: Protein

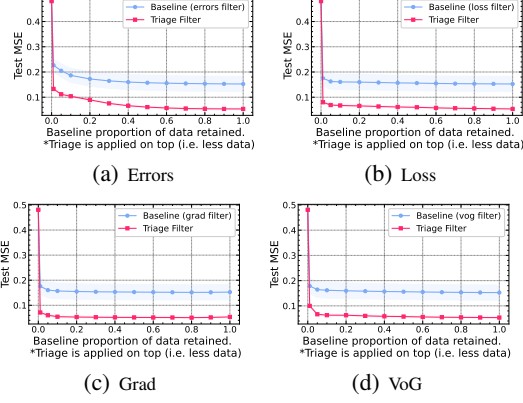

(a) Errors

(b) Loss

(c) Grad

(d) VoG

Figure 28: Fine-grained filter: Star

## C.12    Value of feature collection/acquisition: Additional datasets

**Goal.**    The main paper aimed to assess if TRIAGE could quantify the value of data improvement when we are able to acquire features in Section 5.3.2. We now include results for all datasets (using the same setup, based on the correlation of the feature with the target). Recall we assume that acquiring a valuable feature wrt. the task should increase the proportion of well-estimated samples.

**Takeaway.**    We show the results in Figs 29-37. We see similar results to the main text for all datasets. As we acquire more useful/valuable features, the proportions of well-estimated examples increases as desired. Indicating the measure can capture the value of a feature. We also see that despite the category proportions changing, the metrics themselves remain stable. Indicating that we capture inherent properties to these specific samples, which remain consistent.

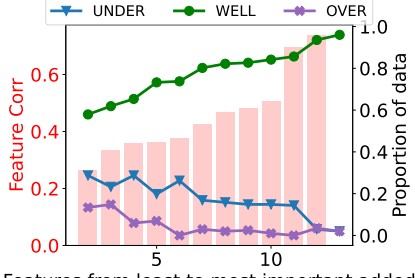

(a) Well estimated subgroup proportion is increased as informative features are acquired.

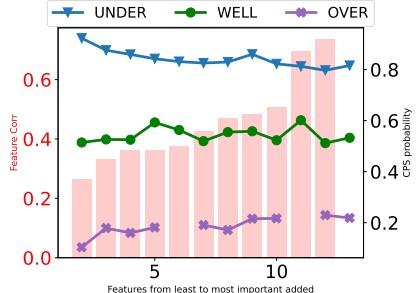

(b) The CPS probabilities for each category remain stable even as the proportions change

Figure 29: Boston dataset

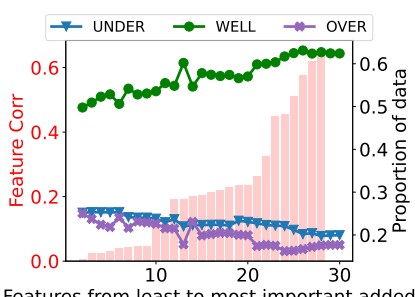

(a) Well estimated subgroup proportion is increased as informative features are acquired.

(b) The CPS probabilities for each category remain stable even as the proportions change

Figure 30: Prostate dataset

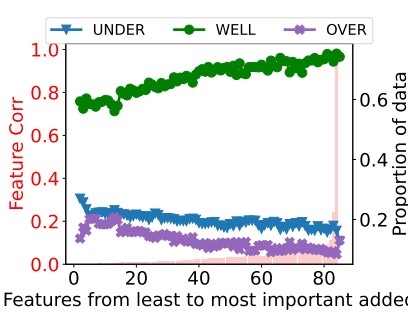

(a) Well estimated subgroup proportion is increased as informative features are acquired.

(b) The CPS probabilities for each category remain stable even as the proportions change

Figure 31: Mimic dataset

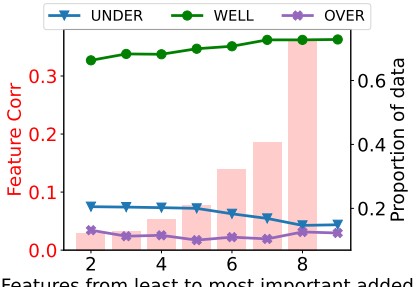
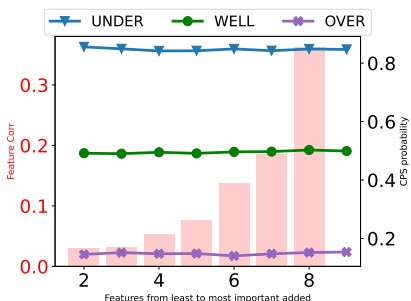

(a) Well estimated subgroup proportion is increased as informative features are acquired.

(b) The CPS probabilities for each category remain stable even as the proportions change

Figure 32: Bio dataset

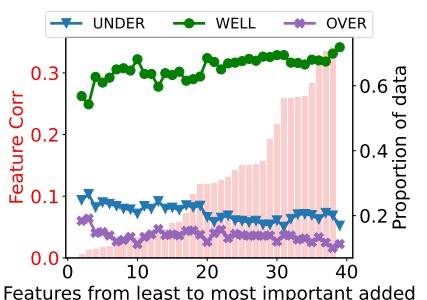
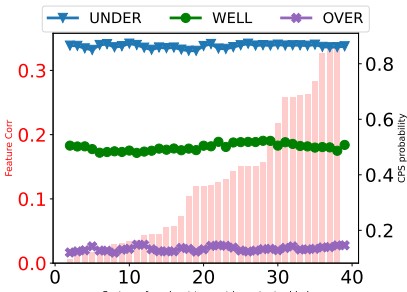

(a) Well estimated subgroup proportion is increased as informative features are acquired.

(b) The CPS probabilities for each category remain stable even as the proportions change

Figure 33: Star dataset

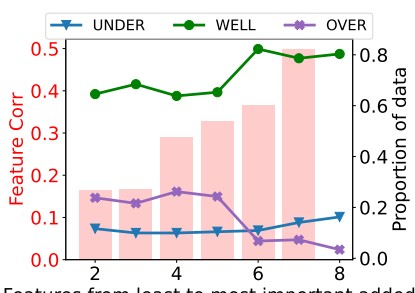
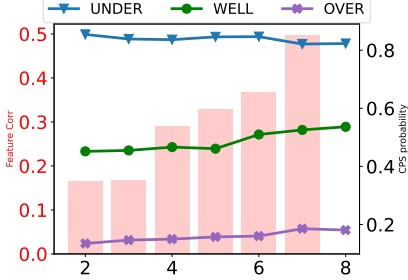

(a) Well estimated subgroup proportion is increased as informative features are acquired.

(b) The CPS probabilities for each category remain stable even as the proportions change

Figure 34: Concrete dataset

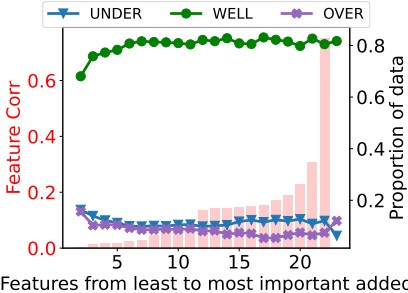
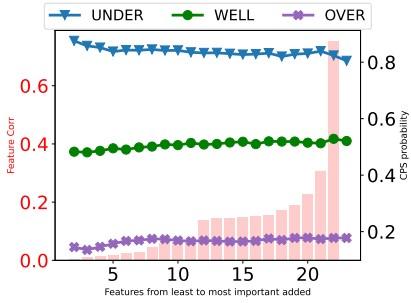

(a) Well estimated subgroup proportion is increased as informative features are acquired.

(b) The CPS probabilities for each category remain stable even as the proportions change

Figure 35: LoS dataset

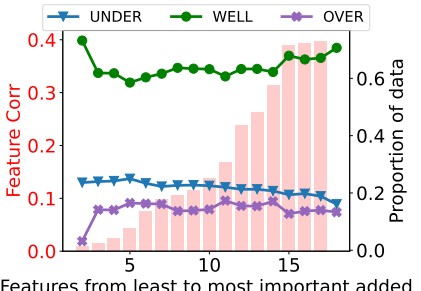
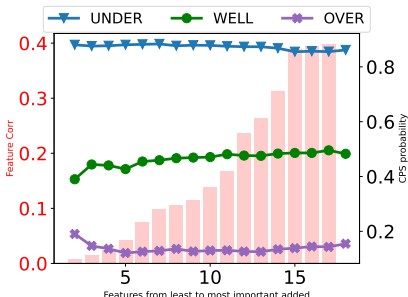

(a) Well estimated subgroup proportion is increased as informative features are acquired.

(b) The CPS probabilities for each category remain stable even as the proportions change

Figure 36: Bike dataset

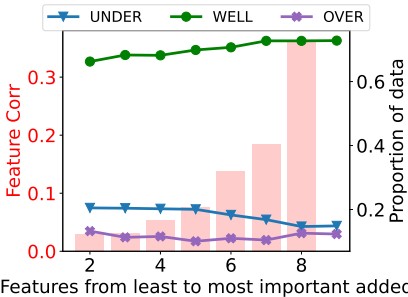
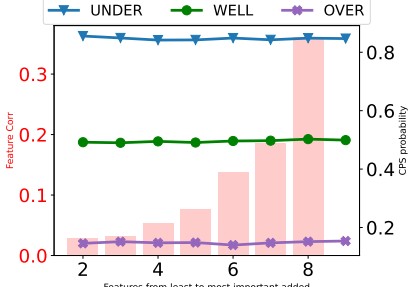

(a) Well estimated subgroup proportion is increased as informative features are acquired.

(b) The CPS probabilities for each category remain stable even as the proportions change

Figure 37: Protein dataset