# OpenReview forum: "TRIAGE: Characterizing and auditing training data for improved regression"
_NeurIPS.cc/2023/Conference — NeurIPS 2023 poster_

### Official Review · Reviewer_N4d8 · 2023-07-03

**Soundness:** 3 good
**Presentation:** 3 good
**Contribution:** 3 good
**Rating:** 6
**Confidence:** 3

**Summary:**

This paper present a data characterization method for regression task. The method leverages conformal predictive systems literature and proposed to estimate training data scores by thresholding percentile of  calibration data points given their conformity measure. The method is interesting in terms of leveraging calibration dataset CPD to measure the training data score. The threshold of grouping (4.3) is less explained, but empirical results show it works. The paper is self-claimed to be the first data characterization method that is suitable for regression task.

**Strengths:**

The proposed method is novel to me. Leveraging calebration dataset to re-score training data is interesting. I am quite curious how the choice of calibration dataset would impact the method's performance.

The paper is very well motivated. And, maybe over motivated, considering justification and explanation missing in the section 4.2 and 4.3.

The experimental results show the proposed method works well in benchmark datasets and could be a good tool to do data selection or feature selection.

**Weaknesses:**

The paper takes pretty long paragraph to highlights its novelty and difference between other data characterization method, which looks like an ads. The motivations part should be reduced and focus more on algorithm explanation. There is very less justification on why thresholding CPD works for the data characterization purpose, and where the rules in Eq 3 come from. They need further justification and explanation.

Algorithm 1 need to further adjustment and state what the "eval sample" is. I think it is one training data in D_train. The output of Algorithm should be a matrix |X| x |M| right? please state it somewhere to help understanding.

Design of 4.2 is not explained. Why mean over training steps? What if we put more weights to higher iteration? What if there are multiple candidate regression methods? would this alter the outcome?

**Questions:**

Why mean over training steps when computing C(x,y)?  What if we put more weights to higher iteration?
What if there are multiple candidate regression methods? would this alter the outcome?
Why does the group assignment threshold designed as Eq 3? Any explanation or justification?

**Limitations:**

No concern. But the authors should describe further in terms of mis using this method, where decision maker can remove data from underrepresented population based on the solution.

---

> ### Author Rebuttal · Authors · 2023-08-09
>
> Dear ``R-N4d8``.
>
> Thank you for your thoughtful comments which have helped improve the paper. We provide answers (A)-(E) & highlight updates to the paper
>
> # (A) Evaluation over all training steps vs looking at higher iterations [Design motivation]
>
> TRIAGE aims to analyze the behavior of different samples through the training process. Specifically, as shown in **Fig. 9 (Appendix A)**, while at the end of training, samples can converge to the same score — the trajectories of TRIAGE scores (training dynamics) are different for different samples. Some samples converge quickly, while others might take longer and oscillate more. We aim to use these differences to delineate samples. As shown in Fig. 9, most of the variability arises in the earlier training steps, which is why we chose to compute the score over all steps to capture this early behavior. While we discuss this in Appendix A.4, we will further update it to be clearer and reference this point in the main manuscript as suggested.
>
> **Experiment:** Beyond explanations, we also thank the reviewer for bringing up this interesting question. It has spurred us to conduct a new experiment to provide additional validation for why we want to evaluate over the entire training trajectory rather than focusing on later/higher iterations.
>
> We specifically have repeated the data sculpting experiment (original results in Table 2). We evaluate computing the TRIAGE score starting at later and later points in the training trajectory after: (i) 33% of training, (ii) 50% of training, (iii) 70% of training and (iv) 80% of training. In contrast to our version in the manuscript, which we will term TRIAGE (Base) is computed from the beginning of training starting at 0% of training). We then compute the TRIAGE scores for each variant (i)-(iv). We then select samples for sculpting and evaluate test MSE for the regressor after sculpting.
>
> We find an increase in MSE (worse performance) the further in the trajectory that we start the score computation. The results averaged over all calibration sizes are shown in **Fig.3 in the uploaded response PDF** and are reflected as MSE increase vs TRIAGE (Base). This highlights the importance of capturing the variability between sample training dynamics early in training before they reach steady state, to differentiate them. We thank the reviewer for the suggestion, as it helps further motivate our use of training dynamics & computing the mean over the whole trajectory.
>
> **UPDATE:** Refine the discussion in Appendix A (referring to it in the main paper) and include the new result in Appendix C.
>
>
> # (B) Multiple candidate regressors
> We evaluate the performance of TRIAGE and specifically the stability of the TRIAGE scores given differently parameterized regressors. Our result in **Fig. 4** highlights that the TRIAGE scores are more consistent compared to baselines — with a Spearman correlation of 0.91. The consistent scores would mean the outcomes of tasks using the scores would be similar — i.e. we have similar outcomes. We have also conducted a similar analysis across different types of regressors in **Appendix C.2 (Table 5)**. We will add a discussion on this in the main manuscript and link it better to bring this result to the fore.
>
> # (C) Adjustments and Clarifications
> Thank you for catching the ambiguity in Algorithm 1. We will update it to be explicit that an evaluation sample is indeed a sample from $D_{train}$. On the output dimensions, yes after all samples are computed, we have an $M$ x $q$ matrix representative of the different CPDs. We then note when we compute the TRIAGE score itself, the dimensionality is $M$ x $E$, where $E$ are epochs.
>
> **UPDATE**: We will update the text with these clarifications and notation updates.
>
> # (D) Streamline Introduction
> Thank you for the suggestion. We will streamline the introduction to make space for further discussion on the algorithm. Specific to your suggestion to include further motivation on the thresholding and curve — this additional space (along with the additional camera ready page) will allow us to bring the algorithmic motivations from Appendix A.5 and Appendix A.6 into the main paper.
>
> # (E) Add discussion on potentially removing under-represented populations
>
> We agree that this is an important consideration, especially around data sculpting when there is a data imbalance between majority and minority groups. We have evaluated this setting in **Appendix C.6.1**. We apologize that this assessment and discussion got lost in the numerous appendices. We will endeavor to better flag it beyond L330-331. Based on your suggestion, we will include it in the discussion Sec.6. To summarize the experimental result in Appendix C.6.1.
>
> We show under this imbalance scenario that TRIAGE still retains strong performance (with the lowest MSE) on the minority group — compared to baselines. This is because of the calibration set containing these a few minority samples, permitting TRIAGE not to filter them and hence retain good performance. This finding highlights an important aspect of calibration set construction to be as representative as possible. We will include a discussion on this as guidance to users of TRIAGE to promote safe and responsible usage.

---

> > ### Comment · Reviewer_N4d8 · 2023-08-15
> > **Thank you**
> >
> > Thank you for the clarification. I would keep my current review with higher confidence.

---

> > > ### Author Response · Authors · 2023-08-16
> > >
> > > Dear ``R-N4d8``
> > >
> > > Thank you for your feedback and suggestions that helped us strengthen the paper!
> > >
> > > Regards
> > >
> > > Paper 11604 Authors

---

### Official Review · Reviewer_pNo8 · 2023-07-06

**Soundness:** 3 good
**Presentation:** 4 excellent
**Contribution:** 3 good
**Rating:** 7
**Confidence:** 2

**Summary:**

The task of data characterization aims to address variations in individual-level performance despite achieving good average performance. Existing methodologies have predominantly focused on classification, leaving a gap in data characterization approaches for regression. In this paper, the authors propose the TRIAGE framework to bridge this gap. Extensive experiments demonstrate the framework's superior performance in regression tasks.


**Strengths:**

- The paper is well-written.

- This is the first paper to introduce a principled data characterization framework in regression settings, supported by extensive experiments.

**Weaknesses:**

Many contents are in the appendix. It would be helpful more discussions about the appendix in the main text.

**Questions:**

- In simplified setting (e.g., linear (kernel) regression?), is it possible to derive further theoretical guarantees on the behavior of C and V, defined in Section 4.2?

- Is the group assignment consistent? (in the sense that under what conditions are the samples accurately assigned their true group label?)

**Limitations:**

refer to weakness

---

> ### Author Rebuttal · Authors · 2023-08-09
>
> Dear ``R-pNo8``.
>
> Thank you for your thoughtful comments which have helped improve the paper. We provide answers (A)-(C) & highlight updates to the paper
>
> # (A) Many appendices - include discussion in the main text
> Thank you for suggesting better flagging our numerous appendices' contents.
>
> **UPDATE:**  In addition to the references interspersed in the text, in the additional camera ready page we will include a section outlining new discussions and results covered in the Appendix.
>
> # (B) Consistency of TRIAGE
> We assess the consistency of the scores computed from the training dynamics for different parameterized models in Fig.4 (Sec 5.1). By virtue of consistency of the scores, this would lead to consistency of group assignment since we use a threshold based mechanism to assign the groups. We find that TRIAGE is much more consistent than the baselines with a Spearman rank correlation or 0.91 averaged across datasets for different parameterized models.
>
> Additionally, in **Appendix C.2**, we further assess consistency using different model classes — namely XGBoost and CatBoost. We apologize that this was unclear linked to your first point about many contents in the Appendix. We will flag this additional set of results in Appendix C.2 in the main manuscript as an additional consistency assessment. The Spearman rank correlation of scores between these different models is similarly high, as shown in Table 5 with a mean of 0.88 +− 0.04. Given the consistency and stability of the scores, this would mean the subsequent assigned groups would be stable. We visualize this in Fig. 13 (Appendix C.2), which illustrates the stability and consistency of the TRIAGE characteristic curves.
>
> **UPDATE:** Flag the additional results in Appendix C.2 around consistency in the main manuscript in Sec 5.1 and in the dedicated discussion on contents in the Appendix.
>
> # (C) Theoretical guarantees
> We agree that providing theoretical guarantees would be valuable beyond just the strong empirical performance of TRIAGE. We wish to highlight two important aspects which make this process challenging:
>
> - Since there is an interdependence of CPD and training dynamics in TRIAGE. Hence, disentangling their impacts theoretically is non-trivial.
>
> - Scores across training epochs are correlated due to iterative model training. This highlights the complexity of any theoretical guarantee, given the dynamic nature of the scores and the correlation of scores. Especially, since independence is often assumed when providing theoretical results — whereas we cannot assume this with a training dynamics based perspective.
>
> **UPDATE:** Given the complexities, we propose to discuss a theoretical guarantee as future work in Sec.6. We will also add an Appendix outlining the aforementioned challenges.

---

> > ### Comment · Reviewer_pNo8 · 2023-08-12
> >
> > I appreciate your thoughtful response. I would prefer to maintain my current score.

---

> > > ### Author Response · Authors · 2023-08-13
> > >
> > > Dear ``R-pNo8``
> > >
> > > Thank you! And thanks again for your time and positive feedback.
> > >
> > > Regards
> > >
> > > Paper 11604 Authors

---

### Official Review · Reviewer_pEXt · 2023-07-07

**Soundness:** 3 good
**Presentation:** 3 good
**Contribution:** 3 good
**Rating:** 6
**Confidence:** 3

**Summary:**

The authors introduce a new data characterization framework, TRIAGE, for regression models. The method utilizes conformal predictive distributions to compute the training examples' scores. To compute TRIAGE scores, the authors use predictive distributions and conformal prediction. A proper training set is used to train a regressor, and a separate calibration set is used for conformal calibration. Conformity measures the dataset's agreement with the observation. Consequently, the conformal predictive scores are computed at each epoch for each training point. Afterward, TRIAGE measures the mean and standard deviation for each training point. The method analyzes each example's training dynamics at each epoch to group each point into one of the groups: under/over/well estimated by the model based on the thresholds. TRIAGE can reduce the number of samples to train the regressor compared with the baseline methods. Specifically, it is observed that the MSE performance is improved with the selected data. It holds an advantage steadily over four different scoring methods.


**Strengths:**

+ The authors propose a novel data characterization method for any regression models that analyze the training dynamic of each training point. Using conformal predictive systems for this problem is intuitive and effective.
+ TRIAGE effectively reduces the number of samples to train regressors compared with the baseline methods. It shows that the MSE performance is improved with data keeping and holds an advantage for four different scoring methods.
+ The paper is well written.


**Weaknesses:**

+ Time complexity can be enormous for large neural networks to compute scores over multiple epochs.
+ Only simple datasets and simple regressors are performed. The data sculpting experiment is also limited to 500 available samples.
+ Some more challenging points might be important in the medical field, but if TRIAGE discards those samples, the model would not learn the critical medical case.



**Questions:**

+ How much does the calibration dataset affect the data sculpting performance?
+ How exactly are distances computed in KNN for residuals of the calibration dataset? Does it use some embedding space?
+ What exactly do authors mean by model-agnostic? It seems that conformity scores are computed on residuals which are based on trained regression model predictions.

---

> ### Author Rebuttal · Authors · 2023-08-09
>
>
> Dear ``R-pEXt``.
>
> Thank you for your thoughtful comments which have helped improve the paper. We provide answers (A)-(E) & highlight updates to the paper
>
> # (A) Computational time
> Thank you for bringing up this point. We agree that analyzing the time cost to compute TRIAGE scores is important. We have run a **new experiment** where we vary the dataset size from *2000-100k* samples and assess the TRIAGE score computational time. Naturally, as the dataset size increases, so does computational time. That being said, even at 100k samples computing the TRIAGE scores for all samples takes <2min, highlighting the time efficiency and capability of TRIAGE to scale to large data sample sizes. We show the results in **Fig. 2 of the response pdf**.
>
> **UPDATE:** Add the new result as a new Appendix, as an important analysis of the time cost of TRIAGE.
>
> # (B) Clarifying data sculpting experiment sample size (“limited to 500 samples”)
> We wish to clarify the sample size (Sec 5.2.2). The limited sample size up to 500 samples is just for the $D_{cal}$ (calibration dataset) in order to demonstrate that we only require a few calibration samples. The training set ($D_{train}$) which TRIAGE audits and sculpts, is the SEER medical dataset from the US which has a large sample size of **20k samples**. We apologize if this was unclear.
>
> **UPDATE:**  We will update Sec 5.2.2 to include the sample size of the SEER dataset to make it clear that we audit and sculpt a large dataset and that small samples are confined to $D_{cal}$.
>
> # (C) Clarifying simple datasets & regressors
> We wish to clarify that the datasets used are not simple but rather real-world datasets. As mentioned on L212-219 & Table 4 (Appendix B.2), our *10 datasets span sample sizes (500-100k) & dimensionality (8-85)*. In addition, the medical datasets used are reflective of actual regression settings. For instance, (i) SEER [25] and CUTRACT are Prostate cancer datasets from US and UK hospitals, (ii) Hospital Length of Stay [27] and (iii) MIMIC Antibiotics [28] are also real-world medical datasets.
>
> We also study a variety of powerful regressors, including neural networks, XGBoost and CatBoost. These regressors are likely to be used in practice on these large and high-dimensional datasets.
>
> **UPDATE:**  We will clarify these two aspects at the start of Sec. 5.
>
> # (D) Discarding critical samples (e.g. medical domain)
> We appreciate the important consideration around data sculpting, especially in medical domains where minority sized groups are often critical samples.  Our evaluation on precisely this issue can be found in **Appendix C.6.1**. We realize that our mention of L330-331 may not have drawn enough attention to it.  To summarize, we show the prostate cancer setting where younger patients are a minority-sized group.  In this imbalance scenario, TRIAGE still retains strong performance on the minority-sized group (with the lowest MSE) compared to baselines. The reason is that the calibration set contains a few minority samples, ensuring they aren't overlooked and discarded by TRIAGE. This underscores the importance of a representative calibration set for capturing critical cases. We will include a discussion on this as guidance to users of TRIAGE to promote safe and responsible usage.
>
> **UPDATE:**  Better flag our experiment in Appendix C.6.1 (on data imbalance) & include a discussion in Sec 6 around constructing a calibration set for safe usage of TRIAGE.
>
> # (E) Clarifications
>
> *(i) Clarifying the KNN in TRIAGE:*
>
> We compute the errors (residuals) for all samples in $D_{cal}$. For sample $x_i$ we estimate its difficulty score ($\sigma$) as the mean absolute errors of $x_i$’s k-nearest neighbors in $D_{cal}$, where K=5
>
>
>
> *(ii) Clarifying the term model-agnostic:*
>
> By stating that Conformal Predictive Distributions offer a "model-agnostic" score, we mean that the score can be computed using any regressor. While the reviewer rightly points out that the core is to compute conformity scores (i.e., residuals), these scores can be derived post-hoc from any regressor, requiring only its output predictions. This "model-agnostic" approach contrasts methods like Bayesian ones, which necessitate specific modeling assumptions, thereby limiting their applicability across different regressors (see L130-132).
>
> **UPDATE:**  Explain in the main manuscript that by model-agnostic, we mean the conformity scores can be computed for any regressor just needing the outputs.
>
>
> *(iii) Effect of the calibration dataset:*
>
> The calibration dataset is important to data sculpting as discussed in point (D). i.e. performance on minority sized groups. We have investigated another aspect **Appendix C.5**: how $D_{cal}$ affects the  “validity/calibration”, especially if we violate the exchangeability assumption.
>
> - Appendix C.5.1 shows we have valid CPDs if exchangeability is satisfied; hence will provide good sculpting.
>
> - Appendix  C.5.2 then shows the effect as $D_{cal}$ gets more non-exchangeable. We find that in the non-exchangeable setting when $D_{cal}$ is small e.g. <300 samples we still empirically achieve coverage and have high quality CPDs based on CRPS score & calibration curves. Moreover, we have good MSE sculpting performance, as shown in Table 2. That said, above 0.3 (> 300 samples) we have sufficient samples that violate exchangeability. This reduces the coverage below 0.9. Interestingly, this matches the change-over point in Table 2, where training directly on $D_{cal}$ will lead to better performance than sculpting. The difference though is not significant, with the MSE not drastically harmed. Additionally, our CPDs are still of high quality, based on the low CRPS score.
>
> **UPDATE:** For clarity, we'll revise our reference on L337-339 to explicitly mention that Appendix C.5 evaluates the effects of $D_{cal}$.

---

> > ### Comment · Reviewer_pEXt · 2023-08-13
> >
> > I appreciate authors for their great effort for responses.
> >
> > I have two small questions:
> >
> > + Why does Figure 16 shows the calibration size only up to 0.5?
> >
> > + I believe the neural network investigated by the authors is the Bayesian NN. What is the architecture of that network and how was it trained?

---

> > > ### Author Response · Authors · 2023-08-14
> > > **Clarifications**
> > >
> > > Dear ``R-pEXt``
> > >
> > > Thank you for your feedback — we are glad our response has helped to address your comments. We clarify your two questions below.
> > >
> > > (1) Figure 16 calibration size: To clarify, this experiment corresponds to the calibration sample sizes of Table 2 in the main paper, where $D_{cal}$ sizes range from 10-500 samples.   For instance, as mentioned on L929:  0.3 (300 samples). In Table 2, we show that at 500 samples (corresponding to 0.5 in Fig 16), that training on $D_{cal}$ directly is better than sculpting $D_{train}$. i.e. TRIAGE sculpting was beneficial at small sample sizes of $D_{cal}$. Thus, we capped the x-axis of Fig 16 to match Table 2. To prevent confusion, we will adjust Figure 16's x-axis to display the raw calibration size, aligning it better with Table 2.
> > >
> > > (2) Neural network clarification: To clarify, the BNN is only used for the baseline “BNN sculpt” --- comparing prediction-based sculpting with uncertainty to TRIAGE sculpting. The BNN is a MLP regressor where we learn a distribution over the weights using variational inference, in the same way as [R1] --- allowing integration of uncertainty. In contrast, when using TRIAGE with an MLP, we *do not* learn a distribution over the weights. Instead, TRIAGE wraps a conventional MLP regressor that uses point estimate weights. The MLP architecture however is the same between the BNN and the TRIAGE MLP.
> > >
> > > We are grateful for the reviewer's time and suggestions, which have strengthened the paper. We hope these clarifications address your questions.
> > >
> > > Paper 11604 Authors
> > >
> > > [R1] Ghosh, Soumya, Jiayu Yao, and Finale Doshi-Velez. “Structured variational learning of Bayesian neural networks with horseshoe priors.” International Conference on Machine Learning. PMLR, 2018.

---

> > > > ### Comment · Reviewer_pEXt · 2023-08-21
> > > >
> > > > Thank you for your clarification.
> > > >
> > > > I will keep my positive score for the paper!

---

> > > > > ### Author Response · Authors · 2023-08-21
> > > > >
> > > > > Dear ``R-pEXt``
> > > > >
> > > > > Thank you! And thanks again for your time, suggestions and positive feedback!
> > > > >
> > > > > Regards
> > > > >
> > > > > Paper 11604 Authors

---

### Official Review · Reviewer_oVcJ · 2023-08-01

**Soundness:** 3 good
**Presentation:** 1 poor
**Contribution:** 2 fair
**Rating:** 6
**Confidence:** 3

**Summary:**

The problem studied in this work is the following: Given a dataset $\lbrace (x_i, y_i ) \rbrace_{i=1}^M$ and a regressor $f_\theta$ trained on this dataset, assign a group label $g_i$ to each sample that specifies whether the regressor under or overestimates on the sample. Such group labels can be used to identify and remove outliers in the dataset. The key formula of the proposed method is Eqn. (1), which is similar to Eqn. (5) in [23] that takes into account the "prediction accuracy" over each sample. The authors proposed to first put each sample into one of the several bins defined on a calibration set, and then assign a CPD score accordingly. The group label is then assigned using the CPD score as well as its variance. In the experiments, the authors use these group labels to remove outliers in the training set, and observe that it can improve the regression performance for a variety of regressors.

**Strengths:**

- The problem studied in this work is very important for real applications since most datasets contain outliers.
- The desiderata for data characterization, as listed in (P1)-(P4) after line 44, are very clear and define the overall goal of this work.
- The experimental results presented in Section 5 support that the proposed method satisfies the desiderata.

**Weaknesses:**

- The overall presentation is not good enough and there are many confusing points in this work, which I will discuss in the Questions section.
- The authors claim TRIAGE to be "the first data characterization framework tailored to regression settings" (lines 63-64), yet in [23] cited in this work, a similar approach for regression conformal prediction was proposed. In fact, the proposed method in this work largely resembles the method in [23], including the CPD score definition Eqn. (1) (versus Eqn. (5) in [23]), the use of calibration scores $\lbrace \alpha_1,\cdots,\alpha_q \rbrace$ (versus Eqn. (1) in [23]), and the use of KNN for estimating $\sigma$ (versus Eqn. (11) in [23]). Thus, I am not sure about the contributions of this work. However, I am sure that "TRIAGE is the first data characterization framework tailored to regression" is an overclaim.
- I feel that the writing is too wordy, especially in Section 5, and the authors use some LaTeX tricks which make the manuscript looks more compact than necessary. I think the authors are able to make Section 5 more concise, so that the main text could easily fit in 9 pages and the paper would have a much better shape.

**Questions:**

1. There are several confusing points in the manuscript that require clarification:
(a) In Algorithm 1, line 3, what do you mean by "nearest neighbor residuals of KNN"? Is it the average distance to the k nearest neighbors?
(b) In Eqn. (11) of [23], the definition of $\sigma$ considers two factors: (i) Whether a sample is close to its k nearest neighbors; (ii) Whether the k nearest neighbors have consistent labels. Does this work use the same definition for $\sigma$? If not, what are the differences?
(c) In the definition of the TRIAGE score $T(x,y,\theta)$ in lines 161-162, what do you mean by "$P(y \le f_\theta(x))$"? What is this probability taken over? If $x, y, f_\theta$ are all deterministic, why should there be a probability?


2. Can the authors make a thorough comparison between this work and [23], including the methodology, the algorithm, the experimental setup and results? This could help clarify the contributions of this work.

3. (Bonus) It would be great to make a theoretical connection between the proposed method and robust statistics. Consider a simple case, where $x$ is 1-dimensional, and the goal is to fit $y=f(x)$ on a dataset $(X,Y)$ with outliers. The dataset could follow Huber's contamination model. In such a scenario, I am curious about whether TRIAGE is provably better than the error baseline (as empirically compared in lines 244-260), and whether there is any guarantee on the performance of the predictor with the outliers removed using TRIAGE. A good starter for robust statistics could be the thesis of Jacob Steinhardt: https://cs.stanford.edu/~jsteinhardt/publications/thesis/paper.pdf. This is a bonus point, but such theoretical analysis could greatly enhance the contributions of this work.

**Post rebuttal note:** I have read the rebuttal and had a discussion with the authors. The authors have addressed most of my questions. I have raised my rating from 4 (original) to 6 (current).

**Limitations:**

Limitations are discussed in Section 6.

**Review summary:** I think this work is interesting overall, and there are some good takeaways. However, the writing is confusing at times, and it also seems to me that a large part of this work is very similar to [23]. I believe many people will find this work useful, but I also believe that this work still needs improvement. So I give a borderline rating to this submission though I rarely give borderline ratings in my reviews. If the authors could address my questions during the rebuttal, I am willing to raise my score to 6 or 7, depending on the quality of the rebuttal.

---

> ### Author Rebuttal · Authors · 2023-08-09
>
> Dear ``R-oVcJ``.
>
> Thank you for your thoughtful comments to improve the paper. We provide answers (A)-(D) & highlight updates to the paper.
>
> # (A) Comparing TRIAGE to Ref [23]
> We discuss differences & similarities to illustrate TRIAGE’s contribution.
>
> **Differences:**
> 1. **Objective:** TRIAGE performs data characterization, scoring samples on their impact on a regressor, enabling data-centric tasks like data sculpting, dataset selection & feature acquisition. In contrast, [23] is conventional conformal prediction for predictive uncertainty estimation.
>
> 2. **Algorithm:** (1) TRIAGE uses conformal predictive distributions (CPDs) providing a full predictive distribution. This contrasts [23]’s prediction intervals, which are less informative than a predictive distribution (see L50-52). (2) TRIAGE’s novelty is studying the training dynamics of scores. In contrast, [23] computes prediction intervals *once* after training, not reflecting dynamic changes, which we show are vital to characterize differences between samples.
>
> 3. **Experiments:** TRIAGE tackles data-centric tasks like data sculpting (Sec 5.2),  dataset selection (Sec 5.3.1) & feature collection/acquisition (Sec 5.3.2). In contrast, [23] evaluates the prediction intervals for predictive uncertainty (coverage & efficiency).
>
> 4. **Results:** While [23] tackles different tasks from TRIAGE, we adapt the CP intervals for sculpting (Sec 5.2.2). Table 2's baseline "*CP Intervals Sculpt*" corresponds to a conformal regressor as in [23]. We will clarify this in the revision. Table 2 shows that porting CP intervals for sculpting are less effective than TRIAGE tailored for this task.
>
> **Similarities:** (i) We clarify that Eq 1 (similar to Eq.5 in [23]) is *NOT* the CPD score but simply a conformity score. The CPD is per Eq 2. We will amend L145-150 to clarify. (ii) we clarify that the similarities noted on conformity score & calibration sets usage are general elements fundamental to  conformal prediction itself & not unique to any method. This is akin to loss functions (conformity scores) & validation sets (calibration sets).
>
> **UPDATE:** Add Appendix discussing differences between TRIAGE & [23].
>
> # (B) Clarifications
> (i) Clarify KNN residuals: We compute the errors for all samples in $D_{cal}$. For sample $x_i$ we estimate its difficulty score $\sigma$ as the mean absolute errors of $x_i$’s k-nearest neighbors in $D_{cal}$, where K=5
>
> (ii) Do we use the same KNN definition as [23]: [23] evaluates multiple variants of the KNN score, including Eq 11 (see Papadopoulos et al. (2011)). To clarify, we use the KNN definition in Sec 3.1 of [23], which we describe in (i)
>
> (iii) Clarifying the probability in the TRIAGE score: The CPD in Eq. 2, denoted $Q(y)$, returns a predictive (probability) distribution. We discuss the necessary condition for this interpretation on L148-150.  Then for a specific value $y$, the function returns the estimated probability $P(Y \leq  y)$, where $Y$ - true target and $y$ - prediction $f_{\theta}(x)$. This score is then computed for the $E$ different checkpoint parameter values.
>
> # (C) Streamline Sec. 5
> Based on your suggestion, we will streamline the writing in Sec. 5, allowing us to expand the Figs side-by-side across the page width.
>
> # (D) Bonus - TRIAGE & Robust Statistics
> We are grateful for the reviewer's suggestion & for sharing the thesis resource.
>
> (i) **TRIAGE vs. Robust Statistics**:
> Motivated by your question, we contrast TRIAGE & Robust Statistics. (1) Post-hoc vs Built-in:  TRIAGE wraps a regressor to detect and sculpt outlier samples, whereas Robust Statistics embeds outlier resilience within the model [R1,R2], e.g. via Huber loss [R2].  (2) Additional data-centric applications: TRIAGE tackles diverse "data-centric AI" tasks, like comparing synthetic data (Sec. 5.3.1) and feature acquisition/collection (Sec. 5.3.2), which is beyond the scope of robust statistics.
>
> (ii) **Theoretical Analysis**:
> Connecting TRIAGE theoretically to robust statistics is an intriguing question. However, we highlight two important challenges:
>
> -	Interdependence of CPD and training dynamics in TRIAGE. Disentangling their impacts theoretically is non-trivial.
>
> -	Scores across training epochs are correlated due to iterative model training. This highlights the complexity of any theoretical guarantee, given the dynamic nature of the scores & their correlation.
>
> Given the complexities, we will discuss the theoretical proof as future work in Sec. 6, citing the provided thesis and the suggested Huber contamination setting. We will also add an Appendix outlining the aforementioned challenges.
>
> As an initial step, we provide simulations.
>
> (iii) **Simulation using Huber's Contamination Model**:
> Our simulation setup mirrors [R3], generating data from a linear model ($y = X\beta + \eta$), with $X\sim U[0,1]$ and $\eta\sim \mathcal{N}(0,1)$. Mimicking Huber's model we contaminate the response $y$ corrupting $\epsilon$ samples, replacing ($\eta_i$) with ($\eta_i + c_i$), where ($c_i$) comes from a different distribution.
> We compare TRIAGE with:
> •	Error baseline
> •	Training with Huber Loss
> •	TRIAGE applied to a Huber Loss trained model.
>
> The results in the **uploaded pdf (Fig 1)** show TRIAGE has lower MSE vs the error baseline as $\epsilon$ rises. TRIAGE is also stable in response to contamination by virtue of the clean calibration set.  Interestingly, combining TRIAGE with a model trained using Huber’s loss proves superior to using either alone, highlighting the compatibility of TRIAGE with robust techniques.
>
> **UPDATE**:
>  - Add discussion about proof to Sec.6, citing the Huber setting & the resource (thesis)[R1]
>
>  - New Appendix to discuss the challenges of a theoretical proof
>
> - New Appendix with the simulation results
>
> [R1] J. Steinhardt. Robust learning: Information theory and algorithms
>
> [R2] R. Maronna, et al. Robust statistics: theory and methods
>
> [R3] M. Chen,, et al. A general decision theory for Huber’s ∈-contamination model

---

> > ### Comment · Reviewer_oVcJ · 2023-08-10
> >
> > Thank you for this rebuttal. Here is a list of changes I suggest the authors make to the paper:
> >
> > - (A): Add this comparison to the paper.
> > - (B): Change line 3 of Algorithm 1 to "the average distance to k-nearest-neighbors".
> > - (C): Improve the writing of Section 5.
> > - (D): Add this discussion to the paper.
> >
> > I have one follow-up question: In rebuttal (B), (iii), I still cannot see on which distribution the probability is taken. Let me pose this question in a clearer way. In line 176 of the submission, you wrote: $C(x^i, y^i)  = \frac{1}{E} \sum_{e=1}^E T(x^i, y^i, \theta _ e) $, which is equal to $\frac{1}{E} \sum_{e=1}^E P(y^i \le f _{\theta _e}(x^i))$. Could you tell me on which distribution is the probability $P(y^i \le f _{\theta _e}(x^i))$ taken over?

---

> > > ### Author Response · Authors · 2023-08-10
> > > **Paper updates & clarification**
> > >
> > > Dear ``R-oVcJ``
> > >
> > > Thank you for your feedback on the rebuttal.
> > >
> > > ----
> > > ### (1) Incorporating changes into the updated paper
> > > We will definitely include your suggested changes which have come from our discussions on (A)-(D). These points will be integrated into the revised paper in the sections identified within the **UPDATE** blocks of our response. Thank you for your help in improving the paper!
> > >
> > > -----
> > >
> > > ### (2) Clarification
> > > To clarify Conformal Predictive Systems output valid cumulative distribution functions, which are termed Conformal Predictive Distributions (CPD).  This is the cumulative probability with respect to a label $y$ , given some $x$ and regressor $f$. With CPDs denoted as $Q$, the conformal p-values get arranged into a probability distribution which has the properties of a CDF — thus essentially becoming probabilities, see [22] for more details. *Appendix A.3* outlines the conditions necessary for $Q$ to be related to a CDF.
> > >
> > > Since the CPD has the properties of a CDF, we use the CPD to estimate probabilities that the true target $y$ is less than or equal to a specified threshold/value. Thus, when you ask about the distribution over which the probability is calculated, it's the CPD that provides the probability estimation.
> > >
> > > To be precise, we evaluate the function $Q$ for a specific $f_{\theta_{e}}(x)$ to get the estimated probability $P(y \leq f_{\theta_{e}}(x))$. We then do this for all $f_{\theta_{e}}$ checkpoints where $e \in E$ to get the trajectory of TRIAGE scores for sample $x$.
> > >
> > > We hope this response clarifies and we will incorporate this more detailed explanation into the revised paper.
> > >
> > > ----
> > >
> > > We are grateful for the reviewer's time and suggestions, which have strengthened the paper. We hope these changes address the reviewer's concerns. If you have any other comments or concerns, please let us know. We would be happy to do our utmost to address them.
> > >
> > > Paper 11604 Authors
> > >
> > > [22] Vladimir Vovk, Ivan Petej, Ilia Nouretdinov, Valery Manokhin, and Alexander Gammerman. Computationally efficient versions of conformal predictive distributions. Neurocomputing, 2020

---

> > > > ### Comment · Reviewer_oVcJ · 2023-08-13
> > > >
> > > > If I understand clearly, in $T(x^i, y^i, \theta)$, $y^i$ is a random variable, and you assume that its distribution follows the CPD, and that is how you estimate $P(y^i \le f _ {\theta _ e} (x^i))$. I suggest the authors make this point very clear.
> > > >
> > > > I don't have any further questions. I will raise my rating to 6.

---

> > > > > ### Author Response · Authors · 2023-08-14
> > > > >
> > > > > Dear ``R-oVcJ``
> > > > >
> > > > > We are glad our response clarified matters and would like to thank you for raising your score!
> > > > >
> > > > > In the revised paper, we will incorporate the more detailed explanation to ensure greater clarity.
> > > > >
> > > > > Thanks again for your time and suggestions.
> > > > >
> > > > > Regards
> > > > >
> > > > > Paper 11604 Authors

---

### Official Review · Reviewer_ckq8 · 2023-08-06

**Soundness:** 3 good
**Presentation:** 4 excellent
**Contribution:** 3 good
**Rating:** 7
**Confidence:** 3

**Summary:**

This paper investigates the problem of training data characterization for regression problems. The authors noted that existing research on training data characterization mostly focuses on classification problems and there remains an absence of research for regression problems. This work proposes TRIAGE, a novel framework designed exclusively for regression settings and applies to a variety of tasks.

TRIAGE leverages conformal predictive distributions to provide a model-agnostic scoring method for evaluating how the model performs on each training sample. This framework also enables analysis into the training dynamics from checkpoints, visualizing the samples being under-/well-/over- estimated by the model during the training process.

The proposed framework is useful for a variety of data curation tasks, such as improving the model performance by throwing poorly-predicted training samples. Beyond sample-wise data analysis, the framework also applies to broader tasks such as dataset selection/feature acquisition. The work validates the proposed method with a range of empirical studies on 10 tableau datasets with 500-100k samples, showcasing its value in data characterization in practical cases.

**Strengths:**

The problem being studied and data-centric perspectives are attractive and important. The work is well-motivated and nicely written with easy-to-follow illustrations.

The narrative of the paper is highly structured, skillful, intriguing and informative. I did enjoy reading this paper. The importance of the work is highlighted. References are substantial and high quality.

Objectives P1-P4 and the comparison table with baselines are clear and precise which can be informative and beneficial for a broader audience. And the improvements in these objectives are later validated in empirical studies, facilitating the verification of this work.

The technical body is clear. The analysis of training dynamics from checkpoints is interesting. Novel insights are provided in the takeaways. The work is practical with many interesting use cases.



**Weaknesses:**

I am combining this part with the questions.

1. Since this work is situated in the context of data-centric AI and also studies data characterization problems, it is a bit unnatural to miss out on the comparisons with data valuation methods.

For example, Shapley-based methods are a common benchmark for the general interpretability of ML applications and also apply to characterizing the effects of training data on regression tasks. Also, I was wondering how model-agnostic data valuation ([1]) pipelines work in this setting.

I would like to see discussions both conceptually and empirically on how those methods perform in the context this paper studies and how they differentiate from the proposed approach.

[1] Lava: Data valuation without pre-specified learning algorithms, ICLR 2023

2. What is the rationality for throwing out data that is over/under-predicted? It could be outliers, but would simply throwing out such data hurt the generalizability of the model or its robustness against distributional shifts, which is prevalent in real-world applications? Do the authors have any empirical study results on such cases?

3. Data selection is an established field of research. This work showcases the proposed approach apply to dataset selection tasks. It would be beneficial if the authors could provide a direct comparison with data selection baselines and visualize their performance.

It won't harm the contribution of this paper if its data selection performance is suboptimal compared to methods designed exclusively for data selection tasks. But I think it is important to know how they compare exactly and benchmark the gap, which would be helpful for future works to compare with or improve over.

4. What is the computational overhead of the proposed framework? Can the authors provide the computation time? How scalable is the proposed framework? What is the largest scale the authors are able to apply it to?

The text in Figure 1 can be made larger.



**Questions:**

See Weaknessnes.

**Limitations:**

No major limitations. Questions for discussion are listed above.

---

> ### Author Rebuttal · Authors · 2023-08-09
>
> Dear ``R-ckq8``.
>
> Thank you for your thoughtful comments to improve the paper. We provide answers (A)-(D) & highlight updates to the paper
>
> # (A) Additional comparisons
> Thanks for suggesting an empirical & conceptual comparison to valuation methods (e.g. Shapley-based & LAVA) to strengthen the results.
>
> First, we clarify that the Gradient Norm (GraNd) baseline [12] from the main paper falls into the data selection literature. Besides showing TRIAGE improves upon GraNd. One key disadvantage of selection via GraNd links to (P1) Consistent characterization, where we show in Fig 4 that the GraNd selector is not consistent across model parameterizations with a Spearman correlation of 0.6 compared to TRIAGE’s 0.91. This means [12] would select different subsets for different models compared to TRIAGE. Additionally, GraNd only applies to neural nets which is less flexible than TRIAGE.
>
> *We now compare to data selection w/ data valuation.*
>
> **(i) Experimentally:** We compare two Shapley valuation methods: (i) TMC-Shapley [R1] & (ii) KNN-Shapley [R2] and LAVA [R3] as suggested, which uses optimal transport. We compare in the context of Table 2- data selection for sculpting. The TRIAGE $D_{cal}$ is used as the validation set for these methods.
>
> Note - TMC-Shapley: was computationally unfeasible for all 20k samples. Hence, we sample 5k and compare it to TRIAGE separately. (ii) KNN-Shapley & LAVA: run on the original 20k samples.
>
> **Performance:** The results are shown in the **response pdf (Table 1 & 2)**. These methods are competitive with TRIAGE. However, TRIAGE tailored to regression outperforms them on downstream MSE performance.
>
> **Computational time:**  We assess compute time vs TRIAGE and show for different sizes of $D_{cal}$ that TRIAGE is more time efficient & unlike Shapley methods, our cost doesn’t increase with the size of the $D_{cal}$. KNN-Shapley is 1-3X & TMC-Shapley 600X more expensive than TRIAGE. We show the results in **Fig. 2 in the response PDF**.
>
> **UPDATE:** Include new results in Table 2 and add a new Appendix to discuss the new computational time experiments.
>
> **(ii) Conceptually:** Differences between valuation methods & TRIAGE are: (a) TRIAGE unlocks other data-centric tasks beyond just data selection, e.g. feature-level acquisition/collection and selection between datasets. These are out of scope for data “sample” selection; (b) Unlike TRIAGE, these methods are not tailored to regression; (c) method differences: computationally Shapley-based methods need to assess multiple sample permutations & need to retrain the model many times. Hence, they struggle to scale to very high-samples sizes (e.g. 100k), unlike TRIAGE where the cost is cheap comparatively (we experimentally compare this later). LAVA uses an additional embedding model to reduce the dimensionality before the optimal transport step.
>
> **UPDATE:** Include a discussion with references on data valuation (Shapley & LAVA) in the related work (Sec. 2).
>
> # (B) Data sculpting under distribution shift
> We agree that the distribution shift setting is interesting to assess.  Our experiment in **Section 5.2.2: "Data sculpting to fit a deployment purpose"** evaluates this setting, where we sculpt $D_{train}$ of US patients to perform well on a different distribution of UK patients at deployment time. We apologize that this was unclear and will update the text to clarify.
>
> Table 2 shows that with a small UK calibration set, TRIAGE sculpts the US data so that the regressor generalizes well on the UK test set, achieving the lowest MSE compared to baselines.
> This highlights the importance of calibration set construction: even in a scenario with a distribution shift, good performance is attainable with TRIAGE by calibrating with a handful of relevant samples.  We will include a discussion on point in Sec. 5.2.2.
>
> Additionally, in **Appendix C.6.1**, we look at generalization for distinct patient groups, specifically the minority sized group. In this imbalanced setup, TRIAGE due to calibration preserves the performance for the minority group, contrasting favorably with baselines. We apologize that this was obscured, given the numerous appendices and will refer to this more prominently than our initial mention on L330-331.
>
> Finally, on insights, we refer the reviewer to **Appendix C.4**, where we show a radial plot (Fig. 14) providing insights into why we sculpt certain samples (over/under-estimated) — thereby allowing us to match our deployment setting better and generalize better. We use the extra camera ready page to add Fig 14 to the main paper to anchor Table 2 with the insights.
>
> **UPDATE:** Improve description of Sec. 5.2.2 to convey this is a distribution shift setting, better flag the results of Appendix C.6.1 in the main paper, which assesses minority and majority sized group performance, move the radial plot (Fig. 14)  to the main paper to provide further insights on sculpting.
>
> # (C) Computational time
> Thank you for the suggestion. We agree that analyzing the time cost to compute TRIAGE scores is important. We have run a new experiment where we vary the dataset size from 2000-100k samples and assess the TRIAGE computational time. Naturally, as the size increases, so does computational time. However, even at 100k samples computing the TRIAGE scores for all 100k takes <2min highlighting TRIAGE’s time efficiency to scale to larger data sizes. We show the results in **Fig. 2(a) of the response pdf uploaded**. We have also compared computational time to valuation methods as mentioned in (A).
>
> **UPDATE:** Add a new Appendix to include the computational time results.
>
> # (D) Text size Fig. 1
> We will increase the font size in Fig. 1 for readability.
>
> [R1] Ghorbani, A., & Zou, J. Data shapley: Equitable valuation of data for machine learning.
>
> [R2] Jia, R et al. Efficient task-specific data valuation for nearest neighbor algorithms.
>
> [R3] Just, H. A et al. LAVA: Data valuation without pre-specified learning algorithms.

---

> > ### Comment · Reviewer_ckq8 · 2023-08-18
> > **Thanks for the rebuttal**
> >
> > I appreciate the authors for their dedicated work and thanks for the response to my comments. My questions have been adequately discussed and I have no further comments at this moment. I would keep my positive rating in support of this work.
> >
> > I hope the authors compile the new results and additional discussions into the paper or its Appendix.
> >
> > Nice work and good luck,
> > Reviewer ckq8

---

> > > ### Author Response · Authors · 2023-08-20
> > >
> > > Dear ``R-ckq8``
> > >
> > > We are glad our response addressed your comments. In the revised paper, we will definitely include these new discussions and results.
> > >
> > > Thanks for your positive feedback and suggestions which have helped us improve the paper!
> > >
> > > Regards
> > >
> > > Paper 11604 Authors

---

### Author Rebuttal · Authors · 2023-08-09

We thank the reviewers for their insightful and positive feedback!

We are encouraged that they found the "problem being studied and data-centric perspectives attractive and important” (**R-ckq8**) for "real applications"  (**R-oVcJ**) and that TRIAGE is a principled (**R-pNo8**) and “novel”  (**R-pEXt,R-N4d8**) data characterization framework in regression settings. Further, they found TRIAGE’s use of conformal predictive systems as "intuitive and effective” (**R-pEXt**) with " analysis of training dynamics from checkpoints is interesting” (**R-ckq8**) — with the "extensive experiments" (**R-pNo8**) supporting the “many interesting use cases”  (**R-ckq8**), such as being a "good tool to do data selection or feature selection” (**R-N4d8**).

We address specific questions and concerns below and highlight updates based on reviewer suggestions that will be incorporated into the revised manuscript.

We have uploaded a ``response pdf`` with additional experiments. These include:

* Simulation study with Huber’s contamination model

* Computational time for 2000-100k samples and compared to Data Valuation methods.

* Motivation for computing the TRIAGE score over all training steps

* Comparison to data valuation methods

On the basis of our clarifications and updates, we hope we have addressed the reviewers' concerns.

Thank you for your kind consideration!

Paper 11604 Authors

---

### Decision · Program_Chairs · 2023-09-21

**Decision:**

Accept (poster)

**Comment:**

This paper studies an important problem to provide a score for each data point which reflects how useful each observation is for the learning algorithm. It does so by using conformal inference ideas (more specifically CPD). This paper was quite well received by the reviewers. While some concerns were raised by the reviewers most of them were answered during the discussion period. The presentation of the paper is pointed out to be limited by reviewer oVcJ. While the paper motivates well their problem the writing is at times sloppy and lacks rigor. Notably, while the algorithm for CPD is well-defined, the definition of the triage score is never clearly defined and stated. However I still recommend acceptance because the proposed method is interesting. However I ask the authors to take the concerns of oVcJ into account to improve the writing.